# LocalFormer: Mitigating Over-Globalising in Transformers on Graphs with Localised Training

**Naganand Yadati**  *y.naganand@gmail.com*
*Independent Researcher*

**Reviewed on OpenReview:**

## Abstract

As Transformers become more popular for graph machine learning, a significant issue has recently been observed. Their global attention mechanisms tend to overemphasize distant vertices, leading to the phenomenon of "over-globalising." This phenomenon often results in the dilution of essential local information, particularly in graphs where local neighbourhoods carry significant predictive power. Existing methods often struggle with rigidity in their local processing, where tightly coupled operations limit flexibility and adaptability in diverse graph structures. Additionally, these methods can overlook critical structural nuances, resulting in an incomplete integration of local and global contexts. This paper addresses these issues by proposing LocalFormer, a novel framework, to effectively localise a transformer model by integrating a distinct local module and a complementary module that integrates global information. The local module focuses on capturing and preserving fine-grained, neighbourhood-specific patterns, ensuring that the model maintains sensitivity to critical local structures. In contrast, the complementary module dynamically integrates broader context without overshadowing the localised information, offering a balanced approach to feature aggregation across different scales of the graph. Through collaborative and warm-up training strategies, these modules work synergistically to mitigate the adverse effects of over-globalising, leading to improved empirical performance. Our experimental results demonstrate the effectiveness of LocalFormer compared to state-of-the-art baselines on vertex-classification tasks.

## 1 Introduction

Graph representation learning Hamilton (2020) enables the extraction of meaningful patterns and relationships from graph-structured data, which is prevalent in many real-world applications such as social networks, biological networks, and transportation systems. Graph Neural Networks (GNNs) Kipf and Welling (2017); Veličković et al. (2018); Hamilton et al. (2017); Xu et al. (2019) effectively extract information from graph data Wu et al. (2022a); Ma and Tang (2020) but struggle with over-smoothing Li et al. (2018) and over-squashing Alon and Yahav (2021), limiting their receptive fields. In contrast, Transformers Vaswani et al. (2017), with their global attention mechanism, offer a promising solution by naturally considering all vertex pairs and adaptively learning interaction relationships from graph data Müller et al. (2024).

The remarkable success of transformers on graphs in graph-level tasks (e.g., molecular property prediction) Kreuzer et al. (2021); Ying et al. (2021); Rampasek et al. (2022a); Wu et al. (2023a) is mainly attributed to their global attention mechanism, which offers enhanced global perception. However, efforts to apply this mechanism to vertex-level tasks have recently revealed the issue of over-globalising Xing et al. (2024), where the attention mechanism disproportionately focuses on higher-order nodes, neglecting more informative lower-order nodes (i.e., local neighbourhoods). Empirical and theoretical analyses indicate that an excessively expanded receptive field can diminish the effectiveness of the global attention mechanism, suggesting the need for a more balanced approach to optimise transformer performance on graphs.

Identifying the weakness of the global attention mechanism of transformers on graphs naturally raises the question of how to improve it to prevent over-globalising while still extracting valuable information from high-order nodes. Integrating a local module, such as GNNs, can alleviate this issue ZHANG et al. (2022); Kong et al. (2023); Liu et al. (2023); Chen et al. (2022); Wu et al. (2021), but the differing properties of local smoothing in GNNs and over-globalising in Graph Transformers complicate the influence on vertex representations. Additionally, the common practice of fusing local and global information through linear combination is inadequate and leads to incorrect predictions Xing et al. (2024), even when either local or global information alone could have been accurate.

To effectively mitigate the over-globalising problem in graph transformers, a balanced approach is crucial: one that integrates a focused local module with a complementary module that focuses on global information *trained and optimised collaboratively*. This collaborative training allows for the preservation of fine-grained, neighbourhood-specific details while simultaneously capturing broader patterns. CoBFormer Xing et al. (2024) adopts this strategy by employing a graph convolutional network (GCN) alongside intra- and inter-cluster transformers to manage local and global information.

Despite these innovations, existing methods face notable limitations. Specifically, the coupled nature of the GCN-based local module restricts flexibility, preventing adaptation to diverse graph structures and limiting the ability to skip irrelevant local information (e.g., less informative one-hop neighbours on heterophilic tasks). To overcome these limitations and address the over-globalising problem, we present the following contributions:

- We introduce LocalFormer, a novel training framework designed to localise a transformer on graphs, and demonstrate that the only existing method is a specific instance of LocalFormer.

- To address the over-globalising issue in transformers on graphs, we explore novel training strategies of LocalFormer, featuring ideas such as collaborative and warm-up training strategies.

- We conduct extensive experimentation on vertex classification datasets to demonstrate the effectiveness of LocalFormer in mitigating over-globalising compared to state-of-the-art baselines.

## 2 Related Work

**Graph Neural Networks (GNNs)** Wu et al. (2022a) are designed to compute vertex representations by recursively aggregating and combining information from neighbouring vertices through a message-passing framework Gilmer et al. (2017). Prominent examples of GNNs include Graph Convolutional Network (GCN) Kipf and Welling (2017), Graph Attention Networks (GAT) Veličković et al. (2018), Graph Sample and Aggregate (GraphSAGE) Hamilton et al. (2017), and Graph Isomorphism Network (GIN)Xu et al. (2019). The issues of over-smoothing Li et al. (2018) and over-squashing Alon and Yahav (2021) hinder GNNs from effectively stacking multiple layers, thereby limiting their ability to capture information from distant vertices. Furthermore, the initial designs of GNNs were based on the homophily assumption Zhu et al. (2020), which posits that connected vertices belong to the same type. Although many GNNs have been designed to handle heterophilic graphs including the most recent methods Wang et al. (2024a); Liang et al. (2024); Wang et al. (2024b); Yu et al. (2024), they continue to encounter challenges such as over-smoothing Park et al. (2024).

**Transformers** Vaswani et al. (2017) utilise global attention mechanisms, effectively constructing fully connected computation graphs with adjustable and learnable edge weights. Extensive research Kreuzer et al. (2021); Ying et al. (2021); Rampasek et al. (2022a) has achieved remarkable success in graph-level tasks, primarily due to their global awareness capability. Building on these, researchers are exploring how to integrate global attention mechanisms into vertex-level tasks Wu et al. (2023a; 2022b); Kong et al. (2023); Liu et al. (2023); Chen et al. (2022); Wu et al. (2021). Despite the benefits, over-globalising can lead to a loss of important local details. Effective strategies are needed to maintain a balance between global and local information Xing et al. (2024). This research precisely focuses on exploring effective approaches, striving to achieve an optimal balance between global and local information. While existing works [1,2,3] demonstrate that proper positional encoding can ensure expressivity and capture specific graph structures, they primarily focus on architectural modifications rather than the training process itself.

**Positioning the Contributions of Our Paper.** While existing transformers, surveyed extensively Müller et al. (2024); Shehzad et al. (2024), demonstrate that proper positional encoding can ensure expressivity and capture specific graph structures, they primarily focus on architectural modifications rather than the training process itself. Our paper introduces CollaborativeLocalFormer (CLF) and WarmLocalFormer (WLF), which systematically mitigate the over-globalising issue in Graph Transformers—an aspect that prior studies have overlooked. Fundamentally, all Graph Transformer architectures inherently suffer from the over-globalising issue due to the nature of the global attention mechanism, which distributes focus across all nodes rather than emphasizing essential local structures Xing et al. (2024). This phenomenon persists regardless of architectural modifications, as global attention mechanisms tend to dilute important local relationships, leading to suboptimal learning. Our primary objective in this paper is to explicitly mitigate this issue through carefully designed training strategies rather than architectural alterations. The superior performance observed in our experimental results is a direct consequence of effectively alleviating this issue, ensuring a more balanced integration of local and global information in Graph Transformers.

## 3 Problem: Over-Globalising of Transformers in Graphs

### 3.1 Notations

We are given an input graph denoted as $\mathcal{G} = (\mathcal{V}, \mathcal{E})$, where the set of vertices $\mathcal{V}$ contains $n$ nodes and the set of edges $\mathcal{E}$ contains $m$ edges. The edges of the graph are encoded by an adjacency matrix $\mathbf{A} = [A_{uv}] \in \{0, 1\}^{n \times n}$, where $A_{uv} = 1$ if there exists an edge from vertex $u$ to $v$, and 0 otherwise. We use $d_v$ to denote the degree of each vertex $v \in \mathcal{V}$, so $d_v = \sum_{u \in \mathcal{V}} A_{vu}$. Often independent of the structural information in the edges of $\mathcal{E}$, we are also given input vertex features encoded in the matrix $\mathbf{X} = [\mathbf{x}_v] \in \mathbb{R}^{n \times d}$, where $\mathbf{x}_v$ is a $d$ dimensional feature vector of vertex $u$.

Let the number of output features for a general task be $c$. In the vertex classification task, vertices are labelled from a label set denoted as $\mathcal{Y}$. Vertex labels are represented with a label matrix $\mathbf{Y} = [\mathbf{y}_u] \in \mathbb{R}^{n \times c}$, where $\mathbf{y}_v$ is the one-hot label of vertex $v$. Let the $c$ classes or labels be $1, \cdots, c$. We denote matrices with bold uppercase letters and vectors with bold lowercase letters.

**Transformers on graphs** allow each vertex in a graph to attend to all other vertices through their global self-attention mechanism as follows:

$$\text{Attn}(\mathbf{H}) = \text{Softmax}\left(\frac{\mathbf{Q}\mathbf{K}^T}{\sqrt{h}}\right)\mathbf{V},$$
$$\mathbf{Q} = \mathbf{H}\mathbf{W}_Q, \mathbf{K} = \mathbf{H}\mathbf{W}_K, \mathbf{V} = \mathbf{H}\mathbf{W}_V, \tag{1}$$

where $\mathbf{H} \in \mathbb{R}^{n \times h}$ denotes the hidden representation matrix and $h$ is the hidden representation dimension. $\mathbf{W}_Q, \mathbf{W}_K, \mathbf{W}_V \in \mathbb{R}^{h \times h}$ are trainable query, key, and value weight matrices of linear projection layers. The attention score matrix is $\hat{\mathbf{A}} = \text{Softmax}\left(\frac{\mathbf{Q}\mathbf{K}^T}{\sqrt{h}}\right) \in [0, 1]^{n \times n}$, containing the attention scores of all vertex pairs. Let $\alpha_{uv}$ be the element of $\hat{\mathbf{A}}$ representing the attention score between vertex $u$ and $v$. Transformers on graphs update vertex representations globally by multiplying the attention score matrix $\hat{\mathbf{A}}$ with the vertex representation matrix $\mathbf{V}$.

### 3.2 Graph Property: k-hop Homophily

While existing homophily metrics Mironov and Prokhorenkova (2024) primarily capture the overall tendency of vertices to share the same label, they fail to provide insights into the varying label similarities across different hops. In the context of over-globalising, *it is crucial to identify at which hop distances neighbours contribute the most useful information for label inference.* To address this, we introduce the k-hop homophily metric, which systematically quantifies the label consistency across different neighborhood ranges. This allows us to visualise and analyse how attention should ideally be distributed in Graph Transformers, ensuring that models do not excessively focus on distant vertices at the cost of local structures.

To demonstrate the over-globalising issue of transformers on graph datasets, we consider two types of graph datasets: heterophilic and homophilic datasets. To measure the homophily level of any type of graph dataset, we adopt the "adjusted homophily" metric of prior work Platonov et al. (2023a;b). It is defined as $\eta = \frac{\eta_{edge} - \sum_{j=1}^{c} \sum_{v:y_v=j} d_v^2/(2|\mathcal{E}|)^2}{1 - \sum_{j=1}^{c} \sum_{v:y_v=j} d_v^2/(2|\mathcal{E}|)^2}$ where $\eta_{edge}$ is the edge homophily defined as $\eta_{edge} = \frac{|\{\{u,v\} \in \mathcal{E}: \ y_u = y_v\}|}{|\mathcal{E}|}$. "Adjusted" homophily is a refined measure that captures the tendency of vertices in a graph to connect with vertices that have the same label, while adjusting for biases such as class imbalances and differences in the number of vertices per class Platonov et al. (2023a;b).

We extend the definition to incorporate vertices in the $k$-hop neighbourhood. We define $d_{v,k}$ as the number of unique vertices exactly $k$-hops away from the vertex $v$ in the graph $\mathcal{G}$ and $\mathcal{E}_k$ as the set of all paths connecting two vertices of length $k$. We propose the $k$-hop homophily as follows:

$$\eta_k = \frac{\eta_{edge,k} - \sum_{j=1}^{c} \sum_{v:y_v=j} d_{v,k}^2/(2|\mathcal{E}_k|)^2}{1 - \sum_{j=1}^{c} \sum_{v:y_v=j} d_{v,k}^2/(2|\mathcal{E}_k|)^2}, \quad \eta_{edge,k} = \frac{|\{\{u,v\} \in \mathcal{E}_k: \ y_u = y_v\}|}{|\mathcal{E}_k|}. \tag{2}$$

Notice that when $k = 1$, we get adjusted homophily, i.e., $\eta_1 = \eta$ and $\eta_{edge,1} = \eta_{edge}$. The proposed metric has the following unique advantages in terms of the interpretation of its value:

- When $\eta_k > 0$, it means that vertices that are $k$ hops away are more likely to share the same label than would be expected by chance.

- $\eta_k = 0$ indicates that there is no specific tendency for vertices at a distance of $k$ to either be similar or dissimilar in terms of their labels.

- When $\eta_k < 0$, it means that vertices that are $k$ hops away exhibit heterophilous tendencies, meaning that vertices at a distance of $k$ are more likely to have different labels than would be expected in a random graph.

**Data-driven Analysis** To clarify the nuanced interpretation of the proposed metric, we analyse several graph datasets. We examine six datasets in this section. Please see the appendix Section A.4 for more datasets. These include three heterophilic datasets: amazon-ratings, minesweeper, and roman-empire Platonov et al. (2023a;b). The remaining three are homophilic datasets: amazon-photo, coauthor-cs Shchur et al. (2018), and wikics Mernyei and Cangea (2020). It is important to note that previous research Xing et al. (2024) has examined over-globalising with respect to homophily using values between 0 and 1 with no specific interpretation for the value of 0.5. In contrast, our proposed $k$-hop homophily metric takes on values between $-1$ and $+1$ with clear interpretations for positive, zero, and negative values.

Figure 1 shows the proposed $k$-hop homophily values on the datasets for varying $k$ from 1 to 9. Firstly, across all datasets, the $k$-hop homophily score shows a downward trend from $k = 2$ onwards, suggesting that vertices farther away are less likely to share the same label. More important, the downward trend levels off at zero, with $\eta_k < 0$ observed only once (in the roman-empire dataset at $k = 1$). This indicates no specific tendency for vertices at a distance of large $k$ to be either similar or dissimilar in terms of their labels.

Interestingly, the $k$-hop homophily values show a peak at $k = 2$ in heterophilic datasets and at $k = 1$ in homophilic datasets. This suggests that in heterophilic datasets, direct neighbours are more likely to have different labels (heterophily), but second neighbours are more likely to have similar labels (homophily). In homophilic graphs, label similarity is strongest locally. Direct neighbours are highly likely to share the same label, but as we move farther, label similarity drops off.

The proposed $k$-hop homophily metric effectively captures the main differences between homophilic and heterophilic graph datasets in this paper, offering a nuanced understanding of label dynamics. Specifically, in homophilic graphs, *the metric exhibits a consistent downward trend from the first hop*, reflecting the diminishing influence of localized label similarity as neighborhoods expand. In contrast, the three heterophilic graphs considered *demonstrate an initial upward trend, followed by a downward trajectory* as demonstrated in Part A of Figure 1.

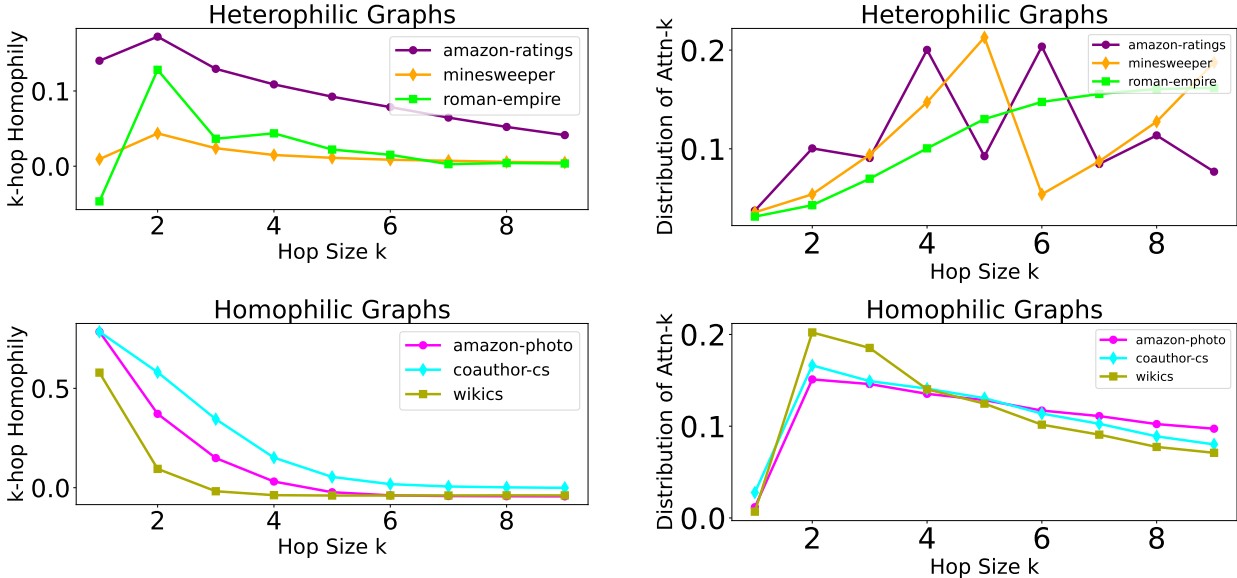

A. k-hop homophily across hop sizes k.       B. Attn-k Distribution in NodeFormer vs. k.

Figure 1: (Best seen in colour) Empirical observations to demonstrate the over-globalising issue in heterophilic and homophilic graphs. The figures show k-hop homophily scores (A) and Attn-k (B) of Node-Former Wu et al. (2022b) for heterophilic (top) and homophilic (bottom) datasets across varying hop sizes k. The plots highlight how homophily decreases with hop sizes, while attention is overly allocated to distant vertices, demonstrating the over-globalising issue. Please see Section 3.2 for details.

### 3.3 Over-Globalising of Transformers on Graphs

In this section, we describe the issue of over-globalising by examining the distribution of attention scores $\alpha_{uv}$ in the matrix $\hat{\mathbf{A}}$ of a well-trained state-of-the-art Transformer on graphs Wu et al. (2022b). We observed that vanilla transformers and most other well-trained transformers on graphs with a globabl self-attention mechansim followed the same trends observed for NodeFormer Wu et al. (2022b). We adopt the average attention score Attn-$k$ of prior work Xing et al. (2024), defined as Attn-$k = \mathbb{E}_{v \in \mathcal{V}} \sum_{u \in \mathcal{N}_{v,k}} \alpha_{vu}$.

Figure 1 shows the visualisation of Attn-$k$ distributions of the well-trained NodeFormer Wu et al. (2022b) on the heterophilic (top right) and homophilic (bottom right) datasets. A higher Attn-$k$ value indicates that the model focuses more on the information from the $k$-th hop. We expect the trends of Attn-$k$ to be roughly similar to those of $k$-hop homophily values (shown correspondingly on the left). However, the situation is different than expected, as the observed results do not align with our expectations.

To begin with, the Attn-$k$ value is lowest at $k = 1$ across all datasets, indicating that there is minimal attention given to the one-hop neighbouring vertices. The observation that Attn-$k$ is nearly zero in homophilic datasets, rises sharply to a peak at $k = 2$, and then shows a slight decline suggests a tendency towards over-globalising when compared to the related $k$-hop homophily values. More importantly, on heterophilic datasets, the variation is much more erratic with values of $k = 5$ (minesweeper), and $k = 6$ (amazon ratings and roman-empire) getting very high Attn-$k$ values, suggesting a much more severe form of over-globalising.

**Why NodeFormer Was Selected.** NodeFormer's design Wu et al. (2022b) represents a state-of-the-art approach to attention mechanisms in graph transformers, offering scalability and efficiency, which makes it a relevant choice for analyzing attention score distribution trends. Although trivial non-graph transformers exhibit over-globalising, it is noteworthy—and not immediately apparent—that even state-of-the-art methods are affected by this issue. Selecting a model that yields unsatisfactory results (please see Tables 1 and 2) on datasets makes over-globalising obvious in a simple attention score visualization, visible to the naked eye.

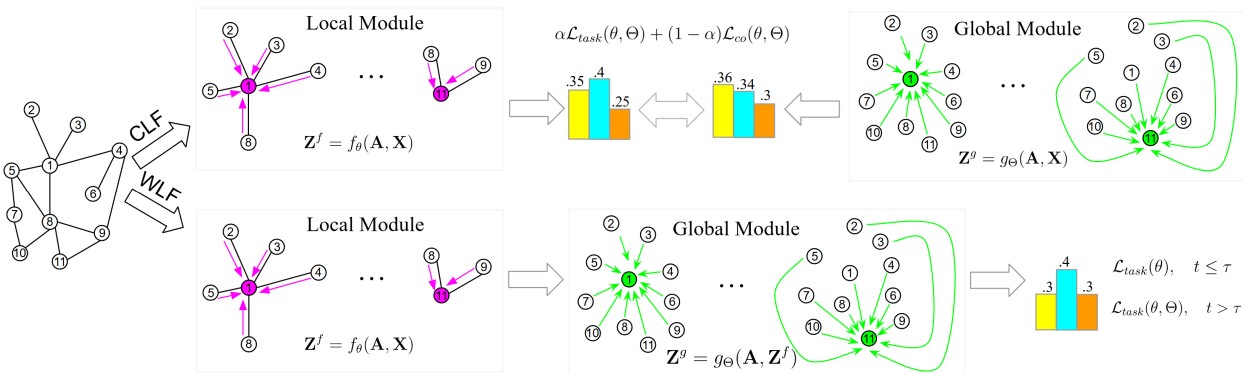

Figure 2: (Best seen in colour) Overview of Collaborative and Warm-up Training Strategies in LocalFormer. In the CollaborativeLocalFormer (CLF) approach, the local and global modules are trained simultaneously, mutually benefiting from each other's learning through collaborative loss $\mathcal{L}_{co}$. In the WarmupLocalFormer (WLF) approach, the local module is trained independently for the initial $\tau$ epochs, followed by the integration of global information from the global module to refine vertex representations, progressively balancing local and global insights. Please see Section 4 for details.

## 4  LocalFormer: Enabling Localised Training in Global Transformers

Considering the findings from the previous section, tackling the over-globalising issue necessitates a balanced strategy that thoughtfully combines both local and global information, ensuring that distant relationships are not given undue priority.

**Two Schemes for Localised Training**

The proposed LocalFormer integrates a local module $f_\theta(\mathbf{A}, \mathbf{X})$ (e.g., sparse attention) to address over-globalising in transformers on graphs. Let $g_\Theta$ represent the global transformer function parameterised by $\Theta$. Unlike the vanilla transformer, which uses only $\mathbf{X}$ as input, graph transformers with a global self-attention mechanism such as NodeFormer Wu et al. (2022b) utilise both the adjacency matrix $\mathbf{A}$ and vertex features $\mathbf{X}$. We denote this global module as $g_\Theta(\mathbf{A}, \mathbf{X})$. It is crucial to note that the local module $f_\theta$, and the global module $g_\Theta$, do not share any information at the hidden layers. Consequently, there is no parameter sharing between $\theta$ and $\Theta$.

The outputs of the local and global modules are integrated at the loss function level, using the collaborative loss $L_{co}$. This integration allows both modules to independently learn complementary representations while contributing to the overall optimisation objective.

Let the output features of $f$ and $g$ be $\mathbf{Z}^f = [\mathbf{z}_v^f]$ and $\mathbf{Z}^g = [\mathbf{z}_v^g]$. We investigate two training schemes to mitigate the over-globalising of $g_\Theta$ by incorporating $f_\theta$. Since $f$ is a local module, we refer to this process as "localised training".

**Scheme 1: Collaborative Training**  In this setup, two models are simultaneously trained on the same training data to improve generalisation capabilities Song and Chai (2018). Within the context of graph machine learning, we train the local $f$ and the global $g$ in a way that they can benefit from each other's learning process. More formally, the collaborative training setup considers two loss functions: a task-specific loss $\mathcal{L}_{task}(\theta, \Theta)$ and a collaborative loss $\mathcal{L}_{co}(\theta, \Theta)$ that is designed to encourage mutual supervision between $f$ and $g$. Mathematically,

$$\mathbf{Z}^f = f_\theta(\mathbf{A}, \mathbf{X}) \in \mathbb{R}^{n \times c}, \quad \mathbf{Z}^g = g_\Theta(\mathbf{A}, \mathbf{X}) \in \mathbb{R}^{n \times c},$$
$$\theta^*, \Theta^* = \arg\min_{\theta, \Theta}\ \alpha\mathcal{L}_{task} + (1-\alpha)\mathcal{L}_{co}. \tag{3}$$

where $\alpha$ is a hyperparameter used to balance the contributions of $\mathcal{L}_{task}$ and $\mathcal{L}_{co}$. We call this training strategy with the local $f$ and the global $g$, "CollaborativeLocalFormer" (CLF).

**Intuition and Details of the Collaborative Loss Function.** Firstly in Equation 3, there is a local module $f_\theta$ and a global module $g_\Theta$ trained collaboritively. Intuitively, the collaborative loss $\mathcal{L}_{co}$ ensures that the two modules communicate with each other by exchanging their expectations about unlabelled data. By considering the confidence or certainty that one module places in its predictions, the other module can reinforce its own learning approach. Mathematically, the loss function $\mathcal{L}_{co}$ can be broken down into two terms as follows:

$$\mathcal{L}_{co}(\theta, \Theta) = \mathcal{L}_{co,g \to f}(\theta, \Theta) + \mathcal{L}_{co,f \to g}(\theta, \Theta)$$

The purpose is to create two-way communication: $f$ learns from $g$ and $g$ learns from $f$. The first term represents a learning mechanism where the local module $f$ is encouraged to align its predictions with those of the global module $g$, *particularly for unlabelled vertices*. Intuitively, this means that module $g$ acts as a teacher, providing soft predictions for vertices where labels are unavailable. The local module $f$ then adjusts its outputs to be more consistent with these predictions. Mathematically, the first term is given by:

$$\mathcal{L}_{co,g \to f}(\theta, \Theta) = -\mathbb{E}_{\mathbf{z}_v^g, v \in \mathcal{V}_U} \log(\mathbf{z}_v^f).$$

Here, $\mathcal{V}_U$ denotes the set of unlabelled vertices and we consider each vertex $v \in \mathcal{V}_U$. The subscript $g \to f$ indicates that module $g$ is guiding module $f$. The term $\mathbf{z}_v^g$ is the soft prediction made by module $g$ for an unlabelled vertex $v$. The term $\log(\mathbf{z}_v^f)$ is the logarithm of the prediction that module $f$ makes for the same vertex. The logarithmic term ensures that the loss is higher when $f$ significantly deviates from $g$, pushing $f$ to refine its understanding in a way that incorporates the global perspective while maintaining local structural awareness. The other term has a similar form as follows:

$$\mathcal{L}_{co,f \to g}(\theta, \Theta) = -\mathbb{E}_{\mathbf{z}_v^f, v \in \mathcal{V}_U} \log(\mathbf{z}_v^g).$$

Intuitively, this term works by having module $g$ adjust its predictions in a way that is consistent with the confidence provided by module $f$. This helps module $g$ refine its predictions based on what module $f$ predicts about the data. The term $L_{\text{task}}$ in Equation 3 represents the primary objective function that ensures both the local module $f_\theta$ and the global module $g_\Theta$ learn to perform the main predictive task effectively. It is typically defined based on supervised learning criteria, such as cross-entropy loss for classification or mean squared error for regression, guiding the models to optimise their outputs towards the ground-truth labels.

**Theorem 4.1.** *CoBFormer Xing et al. (2024) to mitigate over-globalising is an instance of our CLF.*

*Proof.* The specific instantiation is obtained by setting $f_\theta$ to the GCN module Kipf and Welling (2017) and $g_\Theta$ to the Bi-level Global Attention (BGA) module Xing et al. (2024). In the task of vertex classification, the set of vertices $\mathcal{V}$ is partitioned into labelled vertices and unlabelled vertices, so, $\mathcal{V} = \mathcal{V}_L \cup \mathcal{V}_U$. The task-specific loss, $\mathcal{L}_{task}$ is typically cross-entropy, $\mathcal{L}_{task}(\theta, \Theta) = \left(\mathbb{E}_{\mathbf{y}_v, v \in \mathcal{V}_\mathcal{L}} \log(\mathbf{z}_v^f) + \mathbb{E}_{\mathbf{y}_v, v \in \mathcal{V}_\mathcal{L}} \log(\mathbf{z}_v^g)\right)$. The collaborative loss function is designed to encourage mutual supervision between $f$ and $g$ on the unlablled set $\mathcal{V}_U$, given by $\mathcal{L}_{co}(\theta, \Theta) = -\left(\mathbb{E}_{\mathbf{z}_v^g, v \in \mathcal{V}_\mathcal{U}} \log(\mathbf{z}_v^f) + \mathbb{E}_{\mathbf{z}_v^f, v \in \mathcal{V}_\mathcal{U}} \log(\mathbf{z}_v^g)\right)$. □

**Scheme 2: Warm-up Training** This scheme is inspired from techniques that gradually increase the learning rate, starting from a very small value Vaswani et al. (2017); Kalra and Barkeshli (2024). The learning rate slowly reaches a desired level over several iterations or epochs during a "warm-up" phase.

Based on the strong local tendencies of k-hop homophily observed in Figure 1, we propose using only the local module $f_\theta$ for the initial warm-up period of $\tau$ epochs. Afterwards, we refine the vertex representations from the local module $f_\theta$ using $g_\Theta$ in a sequential manner. Mathematically, letting $t$ to be the training epoch

Table 1: Averaged vertex classification results over 10 runs on heterophilic datasets — Accuracy is reported for roman-empire and amazon-ratings, and ROC AUC is reported for minesweeper, tolokers, and questions.We highlight the **first** and the second best results on each dataset.

|  | roman-empire | amazon-ratings | minesweeper | tolokers | questions |
|---|---|---|---|---|---|
| GraphGPS | $82.00 \pm 0.61$ | $53.10 \pm 0.42$ | $90.63 \pm 0.67$ | $83.71 \pm 0.48$ | $71.73 \pm 1.47$ |
| NAGphormer | $74.34 \pm 0.77$ | $51.26 \pm 0.72$ | $84.19 \pm 0.66$ | $78.32 \pm 0.95$ | $68.17 \pm 1.53$ |
| Exphormer | $89.03 \pm 0.37$ | $53.51 \pm 0.46$ | $90.74 \pm 0.53$ | $83.77 \pm 0.78$ | $73.94 \pm 1.06$ |
| NodeFormer | $64.49 \pm 0.73$ | $43.86 \pm 0.35$ | $86.71 \pm 0.88$ | $78.10 \pm 1.03$ | $74.27 \pm 1.46$ |
| DIFFormer | $79.10 \pm 0.32$ | $47.84 \pm 0.65$ | $90.89 \pm 0.58$ | $83.57 \pm 0.68$ | $72.15 \pm 1.31$ |
| GOAT | $71.59 \pm 1.25$ | $44.61 \pm 0.50$ | $81.09 \pm 1.02$ | $83.11 \pm 1.04$ | $75.76 \pm 1.66$ |
| SGFormer | $88.62 \pm 0.50$ | $53.06 \pm 0.29$ | $90.30 \pm 0.28$ | $83.33 \pm 0.68$ | $73.54 \pm 0.65$ |
| CobFormer-G | $88.27 \pm 0.37$ | $52.79 \pm 0.30$ | $89.97 \pm 0.49$ | $83.00 \pm 0.56$ | $73.23 \pm 0.59$ |
| CobFormer-T | $88.56 \pm 0.45$ | $53.04 \pm 0.50$ | $90.30 \pm 0.57$ | $83.36 \pm 0.52$ | $73.48 \pm 0.44$ |
| **CLF (Ours)** | $91.36 \pm 0.39$ | $53.54 \pm 0.24$ | $96.20 \pm 0.68$ | $84.09 \pm 0.40$ | $77.23 \pm 0.66$ |
| **WLF (Ours)** | $\mathbf{91.71 \pm 0.68}$ | $\mathbf{54.02 \pm 0.40}$ | $\mathbf{96.53 \pm 0.64}$ | $\mathbf{84.34 \pm 0.67}$ | $\mathbf{77.52 \pm 0.55}$ |

of the optimisation algorithm and the vertex classification task to be VCT,

$$\mathbf{Z}^f = f_\theta(\mathbf{A}, \mathbf{X}) \in \mathbb{R}^{n \times c}, \quad \mathbf{Z}^g = g_\Theta(\mathbf{A}, \mathbf{Z}^f) \in \mathbb{R}^{n \times c}, \quad \theta^*, \Theta^* = \arg\min_{\theta, \Theta} \mathcal{L}_{task},$$

$$\mathcal{L}_{task} = \begin{cases} \mathcal{L}_{task}(\theta) & \text{if } t \leq \tau \\ \mathcal{L}_{task}(\theta, \Theta) & \text{if } t > \tau \end{cases}, \quad \text{VCT} : \begin{cases} \mathcal{L}_{task}(\theta) = \mathbb{E}_{\mathbf{y}_v, v \in \mathcal{V}_\mathcal{L}} \log(\mathbf{z}_v^f) & \text{if } t \leq \tau \\ \mathcal{L}_{task}(\theta, \Theta) = \mathbb{E}_{\mathbf{y}_v, v \in \mathcal{V}_\mathcal{L}} \log(\mathbf{z}_v^g) & \text{if } t > \tau \end{cases}. \quad (4)$$

We refer to this training strategy as WLF, which stands for WarmLocalFormer with a warm-up period of $\tau$ epochs. Naturally, WLF with $\tau = 0$ corresponds to the typical sequential model where the vertex representations from the local module $f_\theta$ are sequentially refined by $g_\Theta$ to produce the output $\mathbf{Z}^g$.

Please see Appendix Section A.3 for an analysis of computational complexity.

# 5 Experiments

We thoroughly assess LocalFormer to mitigate over-globalising by comparing it with the latest graph transformer models on both homophilic and heterophilic graphs. It is important to note that our primary emphasis in the paper is to mitigate over-globalising and *not to achieve state-of-the-art* on the datasets we evaluated. In fact, because over-globalising is a phenomenon specific to transformers and not to neighbourhood message-passing models, we chose not to include GNN baselines in some of our analysis (e.g., comparison of test accuracy on datasets). Additionally, we conduct ablation studies to evaluate the effectiveness of all the components of LocalFormer. We also analyse the attention score distribution of LocalFormer to demonstrate its capability in mitigating over-globalising.

## 5.1 Performance Comparison with Existing Transformers

We choose a total of ten datasets for our evaluation. Five of them are the homophilic graphs Computer, Photo, CS, Physics Shchur et al. (2018), and WikiCS Mernyei and Cangea (2020). The remaining five are the heterophilic graphs roman-empire, amazon-rarings, minesweeper, tolokers, and questions Platonov et al. (2023a;b). We utilise the public splits from prior work for these datasets. These splits are divided into training, validation, and test sets, maintaining a 50%:25%:25% ratio. Please Appendix Section A.1 for more details on the datasets.

**Baselines.** We compare our proposed methods with eight transformer baselines: GraphGPS Rampasek et al. (2022b), NAGphormer Chen et al. (2023), Exphormer Shirzad et al. (2023), NodeFormer Wu et al. (2022b), DIFFormer Wu et al. (2023b), GOAT Kong et al. (2023), CoBFormer Xing et al. (2024), and SGFormer Wu et al. (2023a). Please see Section A.2 for details on the hyperparameters.

Table 2: Averaged vertex classification accuracy (%) ± std over 10 runs on homophilic datasets. We highlight the  first  and the  second  best results on each dataset.

|  | Computer | Photo | CS | Physics | WikiCS |
|---|---|---|---|---|---|
| GraphGPS | $91.19 \pm 0.54$ | $95.06 \pm 0.13$ | $93.93 \pm 0.12$ | $97.12 \pm 0.19$ | $78.66 \pm 0.49$ |
| NAGphormer | $91.22 \pm 0.14$ | $95.49 \pm 0.11$ | $\mathbf{95.75 \pm 0.09}$ | $\mathbf{97.34 \pm 0.03}$ | $77.16 \pm 0.72$ |
| Exphormer | $91.47 \pm 0.17$ | $95.35 \pm 0.22$ | $94.93 \pm 0.01$ | $96.89 \pm 0.09$ | $78.54 \pm 0.49$ |
| NodeFormer | $86.98 \pm 0.62$ | $93.46 \pm 0.35$ | $95.64 \pm 0.22$ | $96.45 \pm 0.28$ | $74.73 \pm 0.94$ |
| DIFFormer | $91.99 \pm 0.76$ | $95.10 \pm 0.47$ | $94.78 \pm 0.20$ | $96.60 \pm 0.18$ | $73.46 \pm 0.56$ |
| GOAT | $90.96 \pm 0.90$ | $92.96 \pm 1.48$ | $94.21 \pm 0.38$ | $96.24 \pm 0.24$ | $77.00 \pm 0.77$ |
| SGFormer | $91.70 \pm 0.20$ | $94.85 \pm 0.45$ | $94.57 \pm 0.40$ | $96.40 \pm 0.52$ | $73.23 \pm 0.58$ |
| CoBFormer-G | $91.53 \pm 0.24$ | $94.65 \pm 0.22$ | $94.35 \pm 0.40$ | $96.16 \pm 0.54$ | $73.02 \pm 0.64$ |
| CoBFormer-T | $91.81 \pm 0.33$ | $94.61 \pm 0.22$ | $94.39 \pm 0.27$ | $96.57 \pm 0.68$ | $73.97 \pm 0.76$ |
| **CLF (Ours)** | $92.18 \pm 0.29$ | $95.46 \pm 0.20$ | $95.53 \pm 0.16$ | $97.02 \pm 0.18$ | $78.74 \pm 0.47$ |
| **WLF (Ours)** | $\mathbf{92.69 \pm 0.41}$ | $\mathbf{95.63 \pm 0.14}$ | $95.98 \pm 0.35$ | $97.25 \pm 0.15$ | $\mathbf{79.04 \pm 0.44}$ |

We report the best models across hyperparameters as in Section A.2. CLF generates two predictions: one from the local module $f$ and another from the global module $g$. We present the superior result of the two. Please see Appendix 7 and 8 for an analysis of CLF's two modules with different $\alpha$.

**Performance on Heterophilic Graphs.** The empirical results on heterophilic datasets, shown in Table 1, compare the performance of various graph transformer models on vertex classification tasks. The models include the proposed Collaborative LocalFormer (CLF) and Warm-up LocalFormer (WLF). The evaluation metrics are accuracy for roman-empire and amazon-ratings, and ROC AUC scores for minesweeper, tolokers, and questions.

Both CLF and WLF consistently outperform state-of-the-art graph transformer baselines across all heterophilic datasets, with WLF showing the best performance overall, slightly ahead of CLF. This highlights LocalFormer's effectiveness in addressing the over-globalising issue in heterophilic graphs.

The superior performance of CLF and WLF is due to their ability to balance local and global information, avoiding the problem of high attention scores being assigned to distant, often irrelevant nodes in heterophilic graphs.

The $k$-hop homophily plots essentially show that vertices in the 2-hop neighbourhood are the most useful for classification in heterophilic graphs. 2-*hop neighbourhoods are still local* because they retain meaningful label structure around each vertex without requiring full global attention. Our LocalFormer methods outperform previous methods because it dynamically selects the most relevant local information.

**Performance on Homophilic Graphs.** Table 2 shows empirical performance on homophilic datasets. The experimental results demonstrate that LocalFormer effectively addresses the issue of over-globalising in transformers on graphs. The empirical results on homophilic datasets validate the observations made earlier regarding the strong local tendencies in homophilic graphs.

**Experiments on More Non-Homophilic Datasets.** We conduct experiments on more non-homophilic datasets in Appendix Section A.5. The WebKB datasets (Texas, Cornell, and Wisconsin) exhibit fluctuating k-hop homophily trends as shown in Figure 9, with values neither consistently decreasing nor following the heterophilic pattern we previously observed (initial increase followed by a decline). The Wikipedia datasets (Chameleon, Actor, and Squirrel) present intriguing k-hop homophily patterns as shown in Figure 9, with values fluctuating close to zero across different hop sizes.

We selected the three best-performing transformer baselines (Exphormer, SGFormer, CoBFormer) on the initial three heterophilic datasets as baselines for comparison in this new study. We found that our proposed methods (CLF and WLF) performed competitively across all six datasets, though the performance differences were less pronounced than in the three heterophilic datasets initially examined in the paper. Please see Appendix Section A.5 for more details.

## 5.2 Performance-Efficiency Tradeoff Analysis

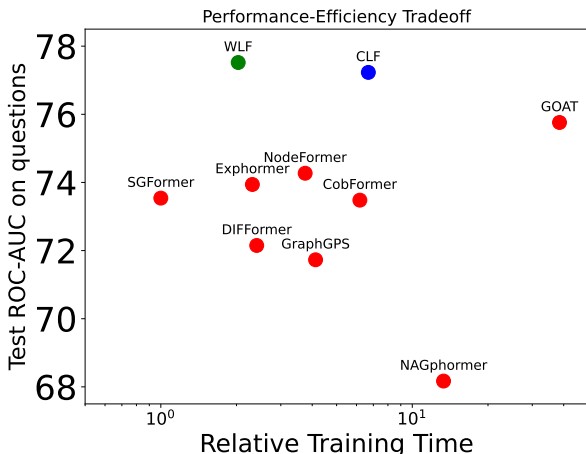

Figure 3: (Best seen in colour) Comparison of ROC-AUC versus relative training time on the questions dataset. Please see Section 5.2 for details.

Figure 3 evaluates the performance of several graph transformer models by plotting test performance (in terms of ROC-AUC) against relative training time on the questions dataset. The aim is to visualise which models strike the best balance between training efficiency and predictive performance. CLF and WLF both achieve a strong balance, with high ROC-AUC scores and moderate relative training times compared to other models. This positions both models as strong candidates for tasks requiring fast yet reliable performance, making them ideal for practical applications where both speed and accuracy are crucial. The visualisation clearly highlights their efficiency and effectiveness in comparison to other existing transformer models.

## 5.3 Ablation Analyses

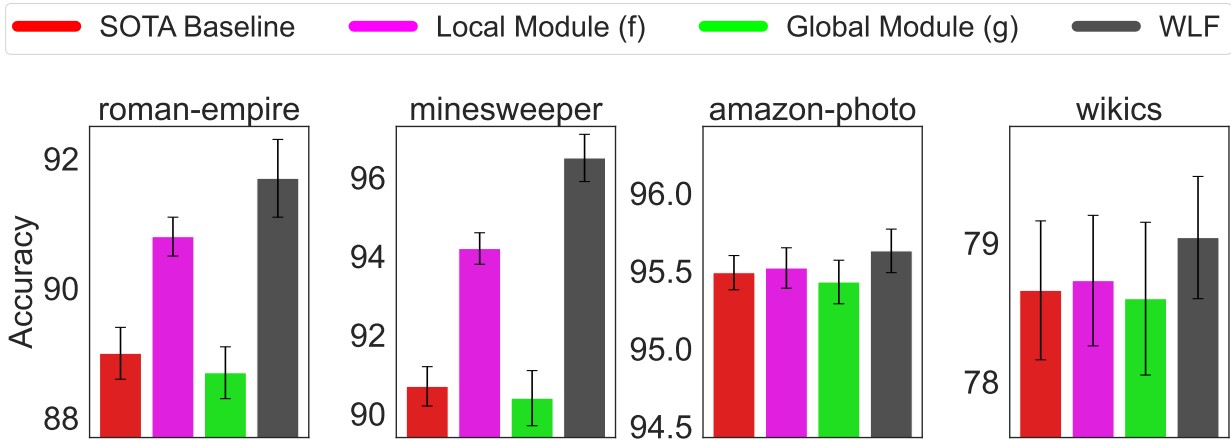

Figure 4: (Best seen in colour) Ablation study on the roman-empire, minesweeper (heterophilic), amazon-photo, and wikics (homophilic) datasets. The results compare the performance of the state-of-the-art (SOTA) baseline, Local Module alone, Global Module alone, and the WarmLocalFormer (WLF) method, showing how integrating both local and global information achieves superior results across different graph types. Please see Section 5.3 for details.

**Ablation of Local and Global Modules**  Figure 4 evaluates the impact of the Local Module, Global Module, and their combination in the WarmLocalFormer (WLF) training strategy on heterophilic (roman-empire, minesweeper) and homophilic (amazon-photo, WikiCS) graphs. This ablation study supports the key claim that over-globalizing in transformers can dilute critical local information. By showing the individual contributions of both modules, it highlights the necessity of a balanced approach. The superior performance of WLF, especially on heterophilic datasets, demonstrates how combining these modules effectively integrates fine-grained, neighborhood-specific patterns with broader graph-wide context, reinforcing the efficacy of LocalFormer. Please see Appendix Figure 7 and Figure 8 for a detailed analysis of the components of CLF.

Figure 7 illustrates scenarios where both local and global modules were trained without $\mathcal{L}_{co}$ in Equation 3. They were trained with the task-specific cross-entropy loss $\mathcal{L}_{task}$. The CLF model, trained with $\alpha\mathcal{L}_{task} + (1-\alpha)\mathcal{L}_{co}$, outperforms these ablated models, highlighting the critical role of $\mathcal{L}_{co}$. Figure 8 examines how the hyperparameter $\alpha$ balances $\mathcal{L}_{co}$ for unlabelled vertices and $\mathcal{L}_{task}$ for labelled vertices. Results indicate that $\alpha = 0.8$ yields optimal performance, with $\alpha = 0.7$ as a close contender.

**Ablation of the Number of Warm-Up Epochs** $\tau$   Figure 5 shows the performance of the WarmLocal-Former (WLF) method across different datasets with varying warm-up epochs ($\tau = 0, 10, 50, 100, 200$) over a fixed 1000 epochs. The warm-up period allows the local module to learn independently before introducing the global module. The study demonstrates that a balanced warm-up period optimises performance by stabilising local features before integrating global information. Shorter warm-ups miss local details, while longer ones can hinder efficient global feature integration.

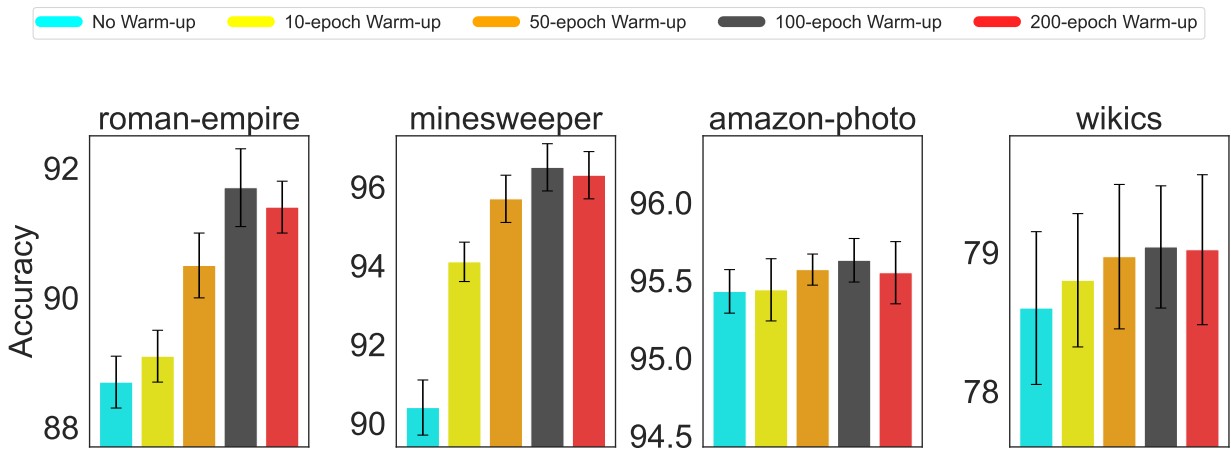

Figure 5: (Best seen in colour) Impact of varying the number of warm-up epochs on the accuracy of the roman-empire, minesweeper, amazon-photo, and WikiCS datasets. Results compare models with no warm-up, $\tau = 10$, $\tau = 50$, $\tau = 100$, and $\tau = 200$ warm-up epochs, illustrating the optimal balance between local and global information at different stages of training. Please see Section 5.3.

## 5.4   Attention Score Distribution of LocalFormer Training Strategies

Figure 6 shows the Attn-$k$ distribution plots for both CLF and WLF, validating the strategies employed by these methods in addressing the over-globalising problem. The collaborative learning in CLF between the local and global modules is evident in the attention distribution, particularly on heterophilic datasets. CLF effectively balances between local and global attention, but it focuses more on semi-distant nodes (3-5 hops), which helps capture cross-cluster relationships crucial for heterophilic graphs. The warm-up strategy of WLF allows the model to first prioritize local information and then gradually integrate global context. The smoother attention distribution across all hop sizes demonstrates the strength of this approach. The attention distributions also show how WLF mitigates the attention spikes observed in existing transformers, offering a more consistent and balanced approach to local-global integration.

The comparison between CLF and WLF highlights their distinct yet complementary strategies in overcoming the over-globalizing challenge in attention mechanisms. By emphasizing semi-distant nodes, CLF enhances the model's ability to capture nuanced relationships across clusters, which is particularly beneficial for heterophilic graphs where nodes differ significantly. In contrast, WLF's gradual integration of global context ensures a smoother transition and a more balanced attention distribution, reducing the potential for attention spikes that can cause instability in the model's focus. These findings imply that leveraging both local and global contexts in a balanced manner can significantly improve the model's performance on complex graph structures by addressing inherent biases and promoting a comprehensive understanding of the data.

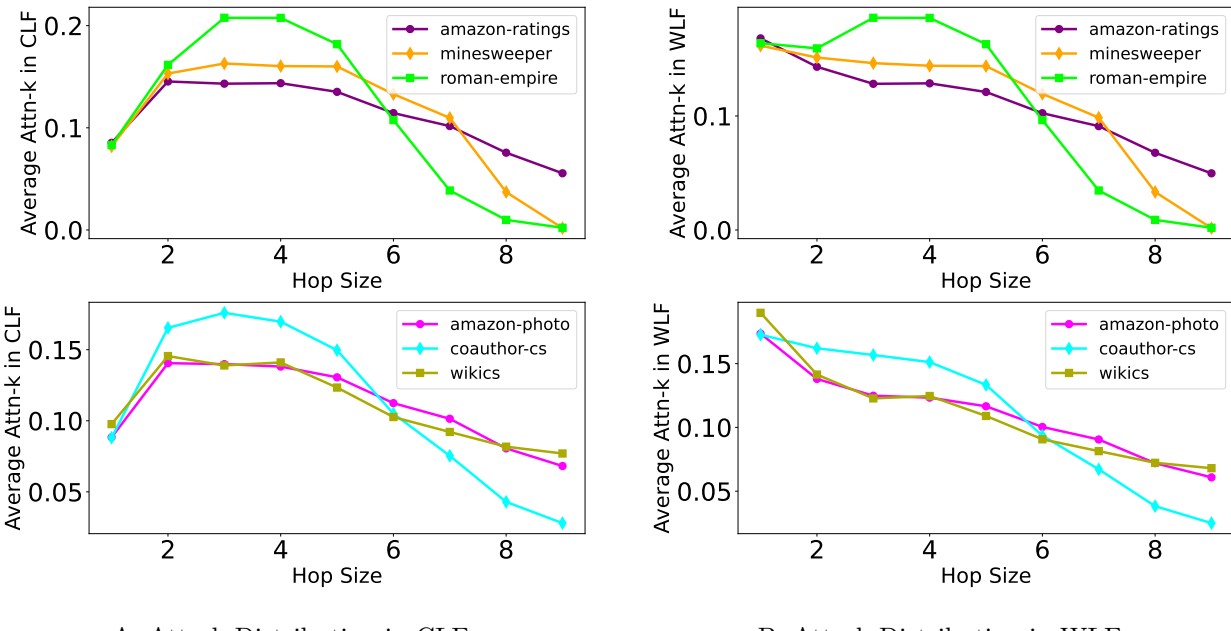

A. Attn-k Distribution in CLF.

B. Attn-k Distribution in WLF.

Figure 6: (Best seen in colour) Average attention distribution of CLF and WLF across different hop sizes $(1-9)$ for heterophilic datasets (amazon-ratings, minesweeper, roman-empire) and homophilic datasets (amazon-photo, coauthor-cs, wikics). Please see Section 5.4 for details.

# 6  Limitations and Future Work

In conclusion, our proposed LocalFormer effectively addresses the issue of over-globalising in graph transformers by introducing two training strategies: collaborative training and warm-up training. This method enhances the selectivity of attention distributions, allowing vertices to prioritise local, relevant information over distant, less pertinent data on both homophilic and heterophilic data. Our paper can be further developed in several directions, particularly by addressing its current limitations.

- **Addressing Structural Dynamics in Graph Data:** Heterophilic and more generally non-homophilic datasets can exhibit counter-intuitive positive homophily values as in Figures 1, 9, and 10, which we believe result from structural factors such as localised homophilic substructures and degree heterogeneity. Investigating these sophisticated dynamics could lead to a refined understanding of graph metrics. This direction warrants further exploration.

- **Theoretical Understanding of Homophily Metrics:** While the paper focuses on overcoming the issue of over-globalising in graph transformers, future research could delve into refining homophily metrics for deeper structural insights. Exploring this comprehensive theoretical analysis can offer a broader perspective on existing measures, informed by recent studies that critically assess and propose new homophily metrics Mironov and Prokhorenkova (2024).

- **Mitigating Over-Globalising Specifically On Heterophilic Graph Datasets:** Future work will explore transformer architectures specifically tailored to heterophilic graphs, particularly focusing on the most informative 2-hop homophily, observed in Figure 1.

- **Scalability:** As transformer models expand to handle larger homophilic Hu et al. (2020) and heterophilic Lim et al. (2021) graphs, scalability becomes crucial. Future work will focus on developing scalable algorithms to mitigate over-globalising, such as sampling-based methods.

- **Metrics for Over-Globalising**: Another key area for future work is the development of quantitative metrics to assess the over-globalising issue in transformers with global self-attention modules.

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

# A  Appendix

## A.1  Dataset Details

Table 3 presents the statistics for all 10 datasets utilised in our experiments. The homophily score is calculated using an existing metric Lim et al. (2021), where a higher score indicates greater homophily.

**Train/Valid/Test splits.** For the *Computer*, *Photo*, *CS*, and *Physics* datasets, we follow the standard practice of randomly splitting the vertices into training (60%), validation (20%), and test (20%) sets Chen et al. (2023); Shirzad et al. (2023). For the other datasets, we use the official splits provided in previous studies Platonov et al. (2023a;b).

Table 3: Statistics of datasets used in our experiments.

| Dataset | Type | Homophily Score | Nodes | Edges | Classes | Features |
|---|---|---|---|---|---|---|
| Computer | Homophily | 0.700 | 13,752 | 245,861 | 10 | 767 |
| Photo | Homophily | 0.772 | 7,650 | 119,081 | 8 | 745 |
| CS | Homophily | 0.755 | 18,333 | 81,894 | 15 | 6,805 |
| Physics | Homophily | 0.847 | 34,493 | 247,962 | 5 | 8,415 |
| WikiCS | Homophily | 0.568 | 11,701 | 216,123 | 10 | 300 |
| roman-empire | Heterophily | 0.023 | 22,662 | 32,927 | 18 | 300 |
| amazon-ratings | Heterophily | 0.127 | 24,492 | 93,050 | 5 | 300 |
| minesweeper | Heterophily | 0.009 | 10,000 | 39,402 | 2 | 7 |
| tolokers | Heterophily | 0.187 | 11,758 | 519,000 | 2 | 10 |
| questions | Heterophily | 0.072 | 48,921 | 153,540 | 2 | 301 |

## A.2  Hyperparameter Details

For all models, including our proposed model and the baseline models, we tune the hyperparameters using a grid search approach. The hyperparameter values that yield the best performance on the validation set are selected.

**GraphGPS Rampasek et al. (2022b).** We choose GAT as the MPNN layer type and Performer as the global attention layer type. We set the number of layers to 2, the number of heads to 8, the hidden dimension to 64, and the number of epochs to 2000. We perform hyperparameter tuning on the learning rate from $\{10^{-4}, 5 \times 10^{-4}, 10^{-3}\}$, and the dropout rate from $\{0.0, 0.1, 0.2, 0.3, 0.4, 0.5\}$.

**NAGphormer Chen et al. (2023).** We set the hidden dimension to 512, the learning rate to 0.001, the batch size to 2000, and the number of epochs to 500. We perform hyperparameter tuning on the number of layers from $\{1, 2, 3\}$, the number of heads from $\{1, 8\}$, the number of hops from $\{3, 7, 10\}$, and the dropout rate from $\{0.0, 0.1, 0.2, 0.3, 0.4, 0.5\}$.

**Exphormer Shirzad et al. (2023).** We choose GAT as the local model and Exphormer as the global model. We set the number of epochs to 2000 and the number of heads to 8. We perform hyperparameter tuning on the learning rate from $\{10^{-4}, 10^{-3}\}$, the number of layers from $\{2, 4\}$, the hidden dimension form $\{64, 80, 96\}$, and the dropout rate from $\{0.0, 0.1, 0.2, 0.3, 0.4, 0.5\}$.

**NodeFormer Wu et al. (2022b).** We set the number of epochs to 2000. Additionally, we perform hyperparameter tuning on the learning rate from $\{10^{-4}, 10^{-3}, 10^{-2}\}$, the number of layers from $\{1, 2, 3\}$, the hidden dimension from $\{32, 64, 128\}$, the number of heads from $\{1, 4\}$, M from $\{30, 50\}$, K from $\{5, 10\}$, rb_order from $\{1, 2\}$, the dropout rate from $\{0.0, 0.3\}$, and the temperature $\tau$ from $\{0.10, 0.15, 0.20, 0.25, 0.30, 0.40, 0.50\}$.

**DIFFormer Wu et al. (2023b).** We use the "simple" kernel type. We perform hyperparameter tuning on the learning rate from $\{10^{-4}, 10^{-3}, 10^{-2}\}$, the number of epochs from $\{500, 2000\}$, the number of layers from $\{2, 3\}$, the hidden dimension form $\{64, 128\}$, the number of heads from $\{1, 8\}$, $\alpha$ from $\{0.1, 0.2, 0.3\}$, and the dropout rate from $\{0.0, 0.1, 0.2, 0.3, 0.4, 0.5\}$.

Table 4: Hyperparameters of LocalFormer per dataset.

|  | Warm-up Epochs | Local-to-Global Epochs | Local Layers | Global Layers | Dropout |
|---|---|---|---|---|---|
| Computer | 200 | 1000 | 5 | 1 | 0.7 |
| Photo | 100 | 1000 | 7 | 2 | 0.7 |
| CS | 100 | 1500 | 5 | 2 | 0.3 |
| Physics | 100 | 1500 | 5 | 4 | 0.5 |
| WikiCS | 100 | 1000 | 7 | 2 | 0.5 |
| roman-empire | 100 | 2500 | 10 | 2 | 0.3 |
| amazon-ratings | 200 | 2500 | 10 | 1 | 0.3 |
| minesweeper | 100 | 2000 | 10 | 3 | 0.3 |
| tolokers | 100 | 800 | 7 | 2 | 0.5 |
| questions | 200 | 1500 | 5 | 3 | 0.2 |

**GOAT Kong et al. (2023).** We set the "conv_type" to "full", the number of layers to 1 (fixed by GOAT), the number of epochs to 200, the number of centroids to 4096, the hidden dimension to 256, the dropout of feed forward layers to 0.5, and the batch size to 1024. We perform hyperparameter tuning on the learning rate from $\{10^{-4}, 10^{-3}, 10^{-2}\}$, the global dimension from $\{128, 256\}$, and the attention dropout rate from $\{0.0, 0.1, 0.2, 0.3, 0.4, 0.5\}$.

**SGFormer Wu et al. (2023a).** We perform tuning on the learning rate in $\{0.001, 0.005, 0.01, 0.05, 0.1\}$, weight decay from $\{10^{-5}, 10^{-4}, 5 \times 10^{-4}, 10^{-3}, 10^{-2}\}$, hidden size within $\{32, 64, 128, 256\}$, dropout within $\{0, 0.2, 0.3, 0.5\}$, and the number of layers from $\{1, 2, 3\}$.

**CoBFormer Xing et al. (2024).** We perform hyperparameter tuning on the learning rate from the set $\{5 \times 10^{-4}, 10^{-3}, 5 \times 10^{-3}, 10^{-2}, 5 \times 10^{-2}\}$, weight decay of the GCN module from $\{10^{-4}, 5 \times 10^{-4}, 10^{-3}, 5 \times 10^{-3}, 10^{-2}\}$, weight decay of the BGA module from $\{10^{-5}, 5 \times 10^{-5}, 10^{-4}, 5 \times 10^{-4}, 10^{-3}\}$, number of clusters within $\{80, 96, 112, 128, 144, 160, 192, 224, 256\}$, the loss balancing parameter $\alpha$ in $\{0.9, 0.8, 0.7\}$ and the temperature $\tau$ from $\{0.9, 0.7, 0.5, 0.3\}$.

**CLF.** The hyperparameters are the same as the CoBFormer model. The main difference is that the local module $f_\theta$ is a 10-layer GAT model with 8 attention heads.

**WLF.** The local module $f_\theta$ contains GAT layers Veličković et al. (2018) with 8 attention heads. The other hyperparameters are shown in Table 4 .

**Justification of the Choices of the Modules.** GAT Veličković et al. (2018) is employed as the local module due to its reliability, while BilevelGlobalAttention (BGA) Xing et al. (2024) is an advanced mechanism designed to extract global information.

These choices are supported by comprehensive real-world experiments and ablation studies in the paper. In particular, Figure 4 explores the effects of training GAT and BGA individually. The results highlight that the novel WarmLocalFormer (WLF) outperforms the use of GAT and BGA trained independently as standalone models. In WLF, a "warm-up" phase is incorporated that focuses exclusively on GAT followed by a sequential phase of refining GAT embeddings with BGA.

Figure 7 shows that CollaborativeLocalFormer (CLF) performs more effectively than GAT and BGA trained independently of each other. CLF also outperforms the CoBFormer baseline [3], which combines GCN and BGA with a collaborative loss function. Experiments on heterophilic (Table 1) and homophilic datasets (Table 2) confirm that both WLF and CLF are more effective than CoBFormer, justifying the choice of using GAT and BGA in LocalFormer.

## A.3 Computational Complexity Analysis.

Given a graph with $n$ vertices and $m$ edges, suppose the hidden dimension is $d \ll n$.

**Complexity of the Local Module**   The local attention module $f_\theta$, which is responsible for local information aggregation, has a complexity of $O(md + nd^2)$. This breakdown can be understood as:

- $O(md)$: This term arises from the need to process each edge in the graph, considering the hidden dimensions. Every edge contributes $d$-dimensional information during aggregation. We assume that the number of hidden layers is a constant (i.e. independent of other variables such as $d, n$).

- $O(nd^2)$: This term is introduced by the feed-forward transformations applied to each vertex in the graph. We assume that the number of input dimensions and the number of output dimensions of the feed-forward layer is at most $d$.

**Complexity of the Global Module**   Without any modifications, a standard transformer's global attention mechanism has a complexity of $O(n^2 d)$, which is quadratic in the number of vertices. This scaling can become prohibitive for large graphs, especially when $n$ is very large. To address the scalability issues, LocalFormer leverages the kernel trick (as used in NodeFormer Wu et al. (2022b)) to linearise the global attention computation. This reduces the global attention complexity to $O(nd^2)$, avoiding the quadratic term with respect to $n$.

- Kernel Trick: This trick approximates the softmax attention, which usually requires computing pairwise attention scores between all nodes, resulting in $O(n^2)$ operations. By applying kernel methods, LocalFormer can bypass the need for pairwise comparisons, achieving a linear approximation of the attention mechanism.

**Complexity of Baselines**   All baseline transformers considered in this work use linear attention mechanisms. Consequently, the complexity is $O(nd^2)$ and hence is not quadratic in $n$.

## A.4   Ablation on CollaborativeLocalFormer (CLF)

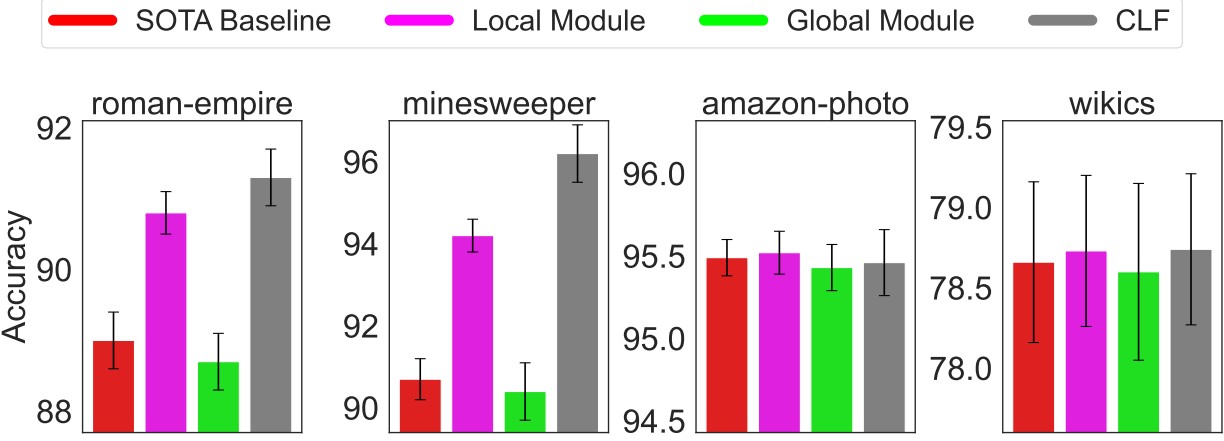

Figure 7: (Best seen in colour) Ablation study comparing the performance of state-of-the-art (SOTA) baseline, Local Module ($f_\theta$), Global Module ($g_\Theta$), and CLF. The local and global modules are trained independently. CLF is trained collaboratively, and we report the better-performing result.

Figure 7 illustrates the ablation study on the local and global modules in CLF. The Local Module achieves high accuracy on homophilic datasets like amazon-photo and wikics, where local information is critical. However, on heterophilic datasets like roman-empire and minesweeper, the Global Module is more competitive, emphasizing the importance of global context. The CLF model outperforms both individual modules across all datasets, highlighting the advantage of combining local and global insights for better overall accuracy. This demonstrates the model's flexibility and robustness in handling diverse data characteristics. Moreover,

the integration of both modules allows the CLF model to adapt dynamically to varying dataset properties, which can be advantageous in real-world applications where dataset homophily and heterophily can fluctuate.

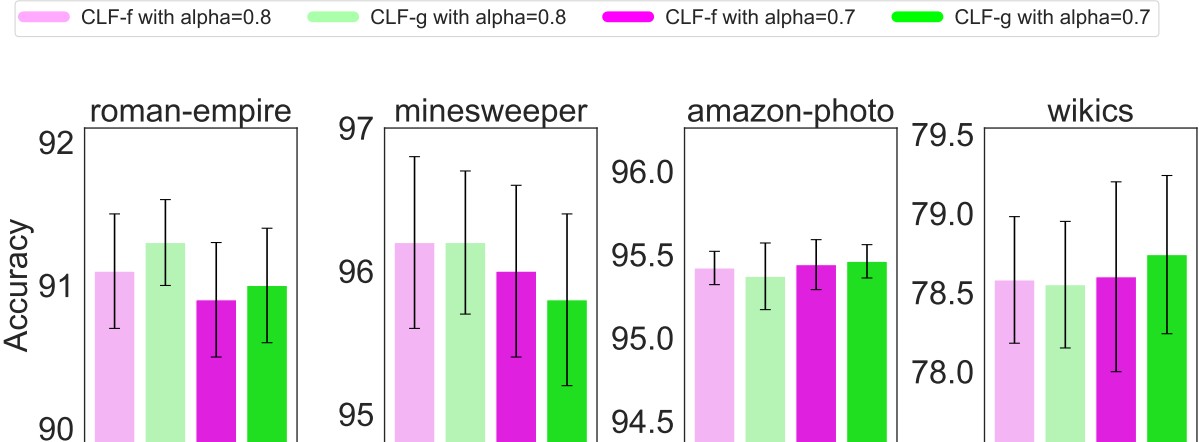

Figure 8: (Best seen in colour) Impact, on test accuracy, of varying the balancing parameter $\alpha$ in $\{0.7, 0.8\}$ on CLF-f (i.e., local information from $f_\theta$) and CLF-g (global information from $g_\Theta$). Note that when two models share the same $\alpha$, they were trained collaboratively.

Figure 8 shows the effect of $\alpha$ for the performance of CLF. For heterophilic datasets like roman-empire and minesweeper, using $\alpha = 0.8$ leads to slightly better performance, suggesting that emphasising the global module is beneficial for these datasets. For homophilic datasets like amazon-photo and wikics, both $\alpha$ values yield similar results. These findings indicate that the parameter $\alpha$ is crucial in tuning the balance between local and global information, helping to optimize the model for specific dataset characteristics. This adaptability underscores the utility of CLF in various contexts, as it can be finely adjusted to leverage the most relevant information type for improved accuracy and performance on a case-by-case basis.

## A.5  Experiments on More Non-Homophilic Datasets

### A.5.1  Experiments on WebKB Datasets

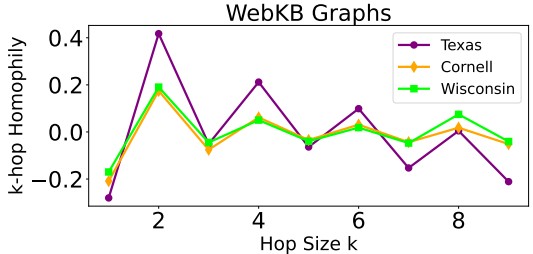

Figure 9: (Best seen in colour) The k-hop homophily metric for WebKB datasets exhibits fluctuating trends.

|  | Texas | Cornell | Wisconsin |
|---|---|---|---|
| Exphormer | **77.78 ± 2.45** | 76.06 ± 3.83 | 77.77 ± 2.62 |
| SGFormer | 76.15 ± 1.99 | 74.97 ± 4.31 | **78.79 ± 2.89** |
| CoBFormer | 76.92 ± 2.51 | 76.02 ± 4.43 | 77.69 ± 1.93 |
| **CLF (Ours)** | 77.32 ± 2.53 | **76.31 ± 3.84** | 78.01 ± 2.03 |
| **WLF (Ours)** | 77.86 ± 2.48 | 76.27 ± 3.91 | **78.62 ± 1.97** |

Table 5: Averaged vertex classification accuracy over 10 runs on WebKB datasets. We highlight the **first** and second best results on each dataset.

In this section, we report experimental results on more datasets.

**WebKB**  A dataset of web pages gathered by Carnegie Mellon University Craven et al. (2000) from the computer science departments of various universities. We focus on three subsets: Cornell, Texas, and Wisconsin. In these subsets, each vertex represents a web page and edges represent hyperlinks between

them. The features of each vertex are represented using the bag-of-words model of the web pages' content. These web pages are manually classified into five categories: student, project, course, staff, and faculty.

**Analysis of k-hop Homophily**  Figure 9 shows k-hop homophily values across varying k on the three WebKB datasets. The k-hop homophily trends for the WebKB datasets (Texas, Cornell, Wisconsin) reveal a complex structural composition, diverging from traditional homophilic and strongly heterophilic behaviours.

Unlike homophilic graphs, where label similarity steadily decreases as neighborhood size expands, these datasets exhibit a more erratic pattern. Specifically, k-hop homophily does not show a monotonic decline but rather fluctuates across different hop distances. This suggests that local label similarity may be preserved within small substructures while connectivity beyond these regions introduces greater label diversity.

Compared to strongly heterophilic graphs, which typically display an initial increase in k-hop homophily followed by a subsequent decline, the WebKB datasets do not consistently follow this pattern. The three WebKB datasets do not exhibit strong homophily, but their label similarity dynamics suggest a *mixed or weakly heterophilic structure.*

**Experimental Results**  We closely follow the experimental setup of a prior work Pei et al. (2020) for training-validation-test split ratios. The classification performance of our proposed methods, CLF and WLF, is evaluated against the three best transaformer baselines (Exphormer, SGFormer, and CoBFormer) on the WebKB datasets (Texas, Cornell, and Wisconsin). The results, presented in Table 5, show that our approaches consistently achieve competitive or superior accuracy across all datasets. The experimental results indicate that WLF is the strongest-performing method across all three WebKB datasets, followed closely by CLF. These results highlight the efficacy of our proposed approaches in handling weak label correlations and complex structural patterns in WebKB graphs.

### A.5.2  Experiments on Wikipedia Datasets

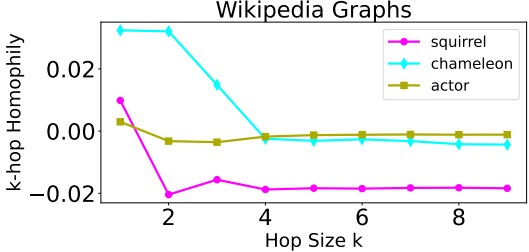

Figure 10: (Best seen in colour) The k-hop homophily metric for Wikipedia datasets fluctuates near zero.

|  | Squirrel | Chameleon | Actor |
|---|---|---|---|
| Exphormer | $65.08 \pm 1.68$ | $66.71 \pm 1.48$ | $36.58 \pm 1.39$ |
| SGFormer | $\mathbf{65.69 \pm 1.87}$ | $66.82 \pm 1.41$ | $\mathbf{37.05 \pm 1.54}$ |
| CobFormer | $64.12 \pm 1.39$ | $66.94 \pm 1.96$ | $36.52 \pm 1.42$ |
| **CLF (Ours)** | $64.18 \pm 1.47$ | $68.06 \pm 1.29$ | $\mathbf{36.97 \pm 1.38}$ |
| **WLF (Ours)** | $\mathbf{65.84 \pm 1.93}$ | $\mathbf{68.92 \pm 1.38}$ | $36.92 \pm 1.46$ |

Table 6: Averaged vertex classification accuracy over 10 runs on Wikipedia datasets. We highlight the first and the second best results on each dataset.

We also conduct experiments on the following datasets:

**Actor**  This dataset represents an actor co-occurrence network. It is derived as the actor-only subgraph from the film-director-actor-writer network Tang et al. (2009). Each vertex represents an actor, and an edge between two vertices indicates that the actors co-appear on the same Wikipedia page. Vertex features are extracted based on certain keywords from these Wikipedia pages. The actors (nodes) are classified into five categories based on the terms found in their Wikipedia entries.

**Squirrel and Chameleon**  These datasets are two page-to-page networks focused on specific Wikipedia topics Rozemberczki et al. (2021). In these datasets, vertices represent individual web pages, while edges represent mutual links between them. The features of each verttex are based on several informative nouns found within the Wikipedia pages. The vertices are categorised into five groups based on the average monthly traffic each page receives.

**Analysis of k-hop Homophily**    The k-hop homophily metric for Wikipedia datasets (squirrel, chameleon, actor) fluctuates near zero, indicating a lack of strong homophilic or heterophilic patterns. Unlike homophilic graphs, the metric does not show a consistent downward trend, nor does it exhibit the expected increase in early hops characteristic of the three heterophilic graphs in Figure 1. These datasets likely contain mixed label interactions with no dominant structural bias.

This behavior indicates that vertices in these graphs do not predominantly connect to either similar or dissimilar labeled vertices, but rather exhibit a more mixed structure. Possible explanations for these trends include high structural diversity, the presence of both homophilic and heterophilic subregions, or degree heterogeneity leading to variations in neighborhood composition. The results suggest that defining these datasets purely as heterophilic may be an oversimplification, and further analysis of their structural properties could provide a clearer understanding of their label connectivity dynamics.

**Experimental Results**    We closely follow the experimental setup of a prior work Pei et al. (2020) for these datasets too. Table 6 illustrates that our methods consistently deliver competitive performance, with WLF achieving the best results across all datasets. These findings highlight the efficacy of our proposed approaches in capturing structural properties, leveraging weak label dependencies, and improving classification performance in non-homophilic graphs. The consistent gains over transformer-based baselines validate the robustness and generalisability of our methods for Wikipedia datasets.

