# OpenReview forum: "LocalFormer: Mitigating Over-Globalising in Transformers on Graphs with Localised Training"
_TMLR — Accepted by TMLR_

### Review · Reviewer_AtT1 · 2024-12-10

**Summary Of Contributions:**

The paper addresses the problem of "over-globalising" in graph transformers, which can lead to suboptimal performance in vertex classification tasks. It provides visualizations to illustrate this issue and proposes a solution that performs better than existing SOTA models.

**Audience:**

Yes

**Claims And Evidence:**

Yes

**Requested Changes:**

See Weakness and Questions.

**Strengths And Weaknesses:**

Strengths:

-The paper visualizes the homophily and attention distribution over multiple hops, effectively highlighting the "over-globalising" issue for graph transformers.

-The proposed model generates competitive results, particularly on heterophilic datasets.

-The Performance-Efficiency Tradeoff Analysis plot offers an intuitive way to evaluate the balance between performance and computational efficiency.

Weakness and Questions:


-The structural integration of a global graph transformer and a local GNN (GAT in the paper) is not clearly explained. It is unclear whether these components share information at each layer or they are only combined at the loss function. Additionally, the mathematical definition of the collaborative loss Lco is not clear.

-Combining a global transformer with a local GNN is not a novel approach. It seems that the collaborative loss Lco is the key factor for the improved performance of the CLF scheme. To validate this, an ablation study of Lco would be highly beneficial. Although Figure 7 compares CLF with the SOTA model, the structure of the SOTA model may be different from the proposed model.

---

> ### Author Response · Authors · 2024-12-11
> **Rebuttal to Reviewer AtT1**
>
> Sincere thanks are extended to the reviewer for noting the strengths of this work.
> The positive comments on the visualisations, the competitive results, and the performance-efficiency tradeoff analysis are appreciated.
> Each highlighted weakness is addressed separately, in a point-wise manner.
>
> $~$
>
> > The structural integration of a global graph transformer and a local GNN (GAT in the paper) is not clearly explained. It is unclear whether these components share information at each layer or they are only combined at the loss function.
>
> The local GNN (GAT), represented by $ f_{\theta} $, and the global transformer, represented by $ g_{\Theta} $, do not share any information at the hidden layers. Consequently, there is no parameter sharing between $\theta$ and $\Theta$.
>
> The outputs of the local and global modules are integrated at the loss function level, using the collaborative loss $L_{co}$. This integration allows both modules to independently learn complementary representations while contributing to the overall optimisation objective.
>
> $~$
>
> > The mathematical definition of the collaborative loss $L_{co}$ is not clear.
>
> Intuitively, the collaborative loss  $L_{co}$ ensures that the two modules communicate with each other by exchanging their expectations about unlabelled data.
> By considering the confidence or certainty that one module places in its predictions, the other module can reinforce its own learning approach.
>
> Mathematically, the loss function $L_{co}$ can be broken down into two terms as follows:
> $$L_{co}(\theta,\Theta)=L_{co,g\rightarrow f}(\theta,\Theta)+L_{co,f\rightarrow g}(\theta,\Theta)$$
> The purpose is to create two-way communication: $f$ learns from $g$ and $g$ learns from $f$. The first term is given by:
>
> $$L_{co,g\rightarrow f}(\theta,\Theta)=-\mathbb{E}_{\mathbf{z}^g_v, v\in\mathcal{V}_U}\log(\mathbf{z}_v^f).$$
>
> * Here, $\mathcal{V}_U$ denotes the set of unlabelled vertices and we consider each vertex $v\in\mathcal{V}_U$.
> * The subscript $g \rightarrow f$ indicates that module $g$ is guiding module $f$.
> * The term $\mathbf{z}^g_v$ is the soft prediction made by module $g$ for an unlabelled vertex $v$.
> * The term $\log(\mathbf{z}_v^f)$ is the logarithm of the prediction that module $f$ makes for the same vertex.
> * Intuitively, this term works by having module $f$ adjust its predictions in a way that is consistent with the confidence provided by module $g$.
> * This helps module $f$ refine its predictions based on what module $g$ predicts about the data.
>
>
> The other term has a similar form as follows:
>
> $$L_{co,f\rightarrow g}(\theta,\Theta)=-\mathbb{E}_{\mathbf{z}^f_v, v\in\mathcal{V}_U}\log(\mathbf{z}_v^g).$$
>
> Here $f$ provides its prediction confidence to inform $g$. Similar to the previous term, this term ensures that module $g$ takes into account insights from module $f$ to enhance its own understanding of the unlabelled vertices.
>
> $~$
>
> > Combining a global transformer with a local GNN is not a novel approach. It seems that the collaborative loss $L_{co}$ is the key factor for the improved performance of the CLF scheme.
>
> While it is true that combining a global transformer with a local GNN has been explored in prior research, this work introduces significant novelties that distinguish it from existing methods:
>
> 1. Two novel training strategies, CollaborativeLocalFormer (CLF) and WarmupLocalFormer (WLF), are proposed to optimise the synergy between local and global modules. The collaborative loss $L_{co}$ promotes mutual enhancement between these components, while the warm-up strategy sequentially refines learning by focusing initially on local patterns before integrating global context.
>
> 2. Graph transformers often suffer from over-globalising, missing finer details. Our LocalFormer framework tackles this issue by balancing detailed, neighbourhood-specific insights from the local module with wide-reaching, global context from the global module. Our experiments validate the effectiveness of this approach, showing strong performance on both homophilic and heterophilic datasets.
>
> $~$
>
> > An ablation study of $L_{co}$ would be highly beneficial. Although Figure 7 compares CLF with the SOTA model, the structure of the SOTA model may be different from the proposed model.
>
> Figures 7 and 8 present comprehensive ablation studies on the collaborative loss $L_{co}$.
>
> Figure 7 illustrates scenarios where both local and global modules were trained *without $L_{co}$*. They were trained with the task-specific cross-entropy loss $L_{ce}$. The CLF model, trained with $\alpha L_{ce} + (1-\alpha) L_{co}$, outperforms these ablated models, highlighting the critical role of $L_{co}$.
>
> Figure 8 examines how the hyperparameter $\alpha$ balances $L_{co}$ for unlabelled verices and $L_{ce}$ for labelled vertices. Results indicate that $\alpha = 0.8$ yields optimal performance, with $\alpha = 0.7$ as a close contender.
>
> $~$
>
> Thanks for the review once again. The review has helped refine the contributions.

---

> > ### Comment · Reviewer_AtT1 · 2024-12-17
> >
> > Thank you for the detailed clarification. I now understand the logic, and I suggest incorporating your explanation into the paper. When I read the paper the first time, I noticed that the definitions of L_task and L_co are defined inside Theorem 4.1. However, I thought these definitions are only valid in the context of this theorem. To enhance clarity, I recommend rearranging the order of the definitions, reasoning, and theorem in the camera-ready version.

---

> > > ### Author Response · Authors · 2024-12-25
> > > **Feedback Appreciated**
> > >
> > > Thanks for the valuable feedback! It is great to see that the explanation has clarified the logic.
> > >
> > > The suggestion to rearrange the order of the definitions for $L_{\text{task}}$ and $L_{\text{co}}$ is duly noted. Presenting these definitions before Theorem 4.1 will indeed help readers understand their broader applicability beyond just the theorem context. This change will be made in the updated version to enhance overall comprehension.

---

### Review · Reviewer_uR7J · 2024-12-23

**Summary Of Contributions:**

This paper addresses the problem of "over-globalization" when applying transformers to graph machine learning, where the models tend to overly focus on higher-order neighborhoods. The authors analyze homophily in single-label graph datasets and provide an example attention mechanism in visualization to support their claims.

To tackle the over-globalization issue, the authors propose LocalFormer, a method that integrates local and global modules. The proposed method can be applied in two training settings: (1) a collaborative training setting where the local and global modules are jointly trained, and (2) a warm-up training setting where only the local module is trained during the initial epochs.

In their experiments, the authors instantiate the local module using GCN and the global module using the BGA module. LocalFormer is evaluated on 10 datasets, demonstrating consistent improvements over SOTA baseline models. Additionally, the authors conduct ablation studies to analyze the effectiveness of the local and global modules, the impact of warm-up epochs on final performance, and the attention scores learned by LocalFormer.

**Audience:**

Yes

**Claims And Evidence:**

Yes

**Requested Changes:**

see above in the weakness.

**Strengths And Weaknesses:**

Strength:
1) The paper is well-written and accessible, making it easy to follow.

2) The claims are well-supported with clear and effective visual evidence.

3) The proposed method is intuitive and straightforward to understand.


Weakness and requested changes:

1) The interpretation of the proposed homophily measure requires stronger justification. It would benefit from a thorough theoretical analysis, including the metric's value range and its interpretation. For instance, while negative values are said as evidence of heterophily in a dataset, only one of the heterophilic datasets (roman-empire) shows negative homophily values in one hop, and even that is near zero. The remaining heterophilic datasets in Figure 1 A all exhibit positive homophily values, which raises questions about the consistency of the metric and reliability in distinguishing heterophilic and homophilic datasets.

2) Provide a clear justification for selecting NodeFormer as the example attention mechanism in Figure 1B to support the generalization of the observations in Section 3. This is particularly important since the experimental results indicate that NodeFormer performs very poorly on both heterophilic and homophilic datasets.

3) LocalFormer is instantiated using GCN and the BGA model without sufficient analysis or justification.

4) Clarify the concept of mutual supervision in the collaborative loss defined in Equation 3. In the proof on page 6, the CO loss is expressed as the sum of expectations of the log values of embeddings of the unlablled nodes learned by the local and global modules. What is the underlying interpretation of this formulation? Does it imply that larger embedding values lead to a smaller CO loss, or is the primary goal to enforce similarity between the embeddings produced by the local and global modules? If the latter, could this enforced similarity contribute to the “over-globalization” issue? Provide a detailed analysis and justification for this design decision.

5) Correct typographical errors, such as the one on page 3 under Equation 1, where "matix" should be corrected to "matrix."

---

> ### Author Response · Authors · 2024-12-25
> **Rebuttal to Reviewer uR7J**
>
> Sincere thanks are expressed to the reviewer for recognising the strengths of this work.
> The positive feedback regarding the clarity, accessibility, and the intuitiveness of the proposed method, as well as the strong visual evidence is valued.
> Each identified weakness will be addressed in a detailed, point-by-point format.
>
> $~$
>
> > The interpretation of the proposed homophily measure requires stronger justification.
>
> The proposed k-hop homophily metric effectively captures the main differences between homophilic and heterophilic graph datasets, offering a nuanced understanding of label dynamics.
> Specifically, in *homophilic graphs*, the metric exhibits a *consistent downward trend* from the first hop, reflecting the diminishing influence of localized label similarity as neighborhoods expand.
> In contrast, *heterophilic graphs* demonstrate an *initial upward trend, followed by a downward trajectory* as demonstrated in Part A of Figure 1 in the paper.
>
> $~$
>
> > For instance, while negative values are said as evidence of heterophily in a dataset, only one of the heterophilic datasets (roman-empire) shows negative homophily values in one hop, and even that is near zero. The remaining heterophilic datasets in Figure 1 A all exhibit positive homophily values, which raises questions about the consistency of the metric and reliability in distinguishing heterophilic and homophilic datasets.
>
> While heterophilic datasets exhibit counter-intuitive positive values, we believe this behavior arises due to structural factors such as localised homophilic substructures and degree heterogeneity, rather than inconsistencies in the metric itself.
> Addressing these sophisticated structural dynamics and refining the metric to account for such nuances is an interesting research direction but falls outside the scope of this paper.
> We will add a discussion of these counter-intuitive observations under the Future Work section.
>
> $~$
>
> >  It would benefit from a thorough theoretical analysis, including the metric's value range and its interpretation.
>
> It is crucial to emphasize that the primary objective of this paper is to tackle the issue of over-globalising in graph transformers, rather than to introduce new homophily measures.
> A comprehensive theoretical analysis refining the proposed homophily metric presents a promising direction for future research.
> This point will also be emphasized in the Future Work section, particularly in light of recent studies [1] that explore various homophily metrics proposed in the literature.
>
>
> $~$
>
> > Provide a clear justification for selecting NodeFormer as the example attention mechanism...This is particularly important since the experimental results indicate that NodeFormer performs very poorly on both heterophilic and homophilic datasets.
>
> NodeFormer's design [2] represents a state-of-the-art approach to attention mechanisms in graph transformers, offering scalability and efficiency, which makes it a relevant choice for analyzing attention score distribution trends.
> Although trivial non-graph transformers exhibit over-globalising, it is noteworthy—and not immediately apparent—that even state-of-the-art methods are affected by this issue.
> Selecting a model that yields unsatisfactory results on datasets makes the issue of over-globalising obvious in a simple attention score visualization, visible to the naked eye.
>
> $~$
>
> > LocalFormer is instantiated using GCN and the BGA model without sufficient analysis or justification.
>
> It is crucial to clarify that LocalFormer is instantiated using Graph Attention (GAT) and Bi-level Global Attention (BGA) [3] as detailed in the Appendix Section A.2, titled "Hyperparameter Details". GAT is employed as the local module due to its reliability, while BGA is an advanced mechanism designed to extract global information. These choices are supported by comprehensive real-world experiments and ablation studies in the paper.
>
> In particular, Figure 4 explores the effects of training GAT and BGA individually. The results highlight that the novel WarmLocalFormer (WLF) outperforms the use of GAT and BGA trained independently as standalone models. In WLF, a "warm-up" phase is incorporated that focuses exclusively on GAT followed by a sequential phase of refining GAT embeddings with BGA.
>
> Figure 7 shows that CollaborativeLocalFormer (CLF) performs more effectively than GAT and BGA trained independently of each other. CLF also outperforms the CoBFormer baseline [3], which combines GCN and BGA with a collaborative loss function. Experiments on heterophilic (Table 1) and homophilic datasets (Table 2) confirm that both WLF and CLF are more effective than CoBFormer, justifying the choice of using GAT and BGA in LocalFormer.
>
> $~$
>
> References:
> 1. Revisiting Graph Homophily Measures, In LoG'24,
> 2. NodeFormer: A Scalable Graph Structure Learning Transformer for Node Classification, In NeurIPS'22,
> 3. Less is More: on the Over-Globalizing Problem in Graph Transformers, In ICML'24.

---

> > ### Author Response · Authors · 2024-12-25
> > **Rebuttal to Reviewer uR7J Continued**
> >
> > > Clarify the concept of mutual supervision in the collaborative loss defined in Equation 3. In the proof on page 6, the CO loss is expressed as the sum of expectations of the log values of embeddings of the unlablled nodes learned by the local and global modules.
> >
> > Firstly, there is a local module $f_{\theta}$ and a global module $g_{\Theta}$ trained collaboritively in the CLF mechanism.
> > Intuitively, the collaborative loss  $L_{co}$ in Equation 3 ensures that the two modules communicate with each other by exchanging their expectations about unlabelled data.
> > By considering the confidence or certainty that one module places in its predictions, the other module can reinforce its own learning approach.
> >
> > Mathematically, the loss function $L_{co}$ can be broken down into two terms as follows:
> > $$L_{co}(\theta,\Theta)=L_{co,g\rightarrow f}(\theta,\Theta)+L_{co,f\rightarrow g}(\theta,\Theta)$$
> > The purpose is to create two-way communication: $f$ learns from $g$ and $g$ learns from $f$. The first term is given by:
> >
> > $$L_{co,g\rightarrow f}(\theta,\Theta)=-\mathbb{E}_{\mathbf{z}^g_v, v\in\mathcal{V}_U}\log(\mathbf{z}_v^f).$$
> >
> > * Here, $\mathcal{V}_U$ denotes the set of unlabelled vertices and we consider each vertex $v\in\mathcal{V}_U$.
> > * The subscript $g \rightarrow f$ indicates that module $g$ is guiding module $f$.
> > * The term $\mathbf{z}^g_v$ is the soft prediction made by module $g$ for an unlabelled vertex $v$.
> > * The term $\log(\mathbf{z}_v^f)$ is the logarithm of the prediction that module $f$ makes for the same vertex.
> > * Intuitively, this term works by having module $f$ adjust its predictions in a way that is consistent with the confidence provided by module $g$.
> > * This helps module $f$ refine its predictions based on what module $g$ predicts about the data.
> >
> >
> > The other term has a similar form as follows:
> >
> > $$L_{co,f\rightarrow g}(\theta,\Theta)=-\mathbb{E}_{\mathbf{z}^f_v, v\in\mathcal{V}_U}\log(\mathbf{z}_v^g).$$
> >
> > Here $f$ provides its prediction confidence to inform $g$. Similar to the previous term, this term ensures that module $g$ takes into account insights from module $f$ to enhance its own understanding of the unlabelled vertices.
> >
> > $~$
> >
> > > Correct typographical errors, such as the one on page 3 under Equation 1, where "matix" should be corrected to "matrix."
> >
> > Thanks for pointing these out. They will be fixed in an updated version of the paper.
> >
> > Thanks for the review once again. The review has helped refine the contributions.

---

> > ### Comment · Reviewer_uR7J · 2025-02-28
> > **Thanks for the reply**
> >
> > Thanks for the clarification. The added discussion in section 3.2 provides clearer interpretation of the metric. I suggest to add short motivation explaining the proposal of the metric in the main paper to help the readers to understand the scope of the current research. The metric is mainly used to visualize in which hop neighborhood are the neighbors most useful to infer the labels of the ego nodes. The added clarification of the choice of NodeFormer and explaination of the loss function in section 4 look good.

---

> > > ### Author Response · Authors · 2025-03-01
> > > **Suggestion Incorporated**
> > >
> > > Thanks for the suggestion.
> > > The submission has been updated, incorporating the suggestion.
> > > Specifically, a new motivation section has been added at the beginning of section 3.2, titled 'Graph Property: k-hop Homophily', right before introducing the formal definition of the metric.

---

### Review · Reviewer_mRfw · 2025-01-23

**Summary Of Contributions:**

This paper studies the problem of over-globalising in Transformers for graphs. The authors propose LocalFormer, a novel framework that includes collaborative and warm-up training strategies, to integrate local information into global computation. Empirical experiment results justify the effectiveness of the proposed method.

**Audience:**

Yes

**Claims And Evidence:**

Yes

**Requested Changes:**

1. It is still not clear why the proposed method works better than existing methods, especially on heterophilic graphs. I think the proposed method emphasizes local graph information, which should be contradictory to a successful learning of a heterophilic graph.

2. I am wondering what the performance of the proposed method is on some simple and basic heterophilic graphs, such as Chameleon, Actor, Squirrel, Wisconsin, etc., used in Geom-GCN.

3. I think your paper may claim an interesting thing, i.e., the training scheme beyond the architecture itself can further benefit graph learning with Transformers. Existing works [1,2,3] prove that a proper positional encoding is expressive enough or guaranteed to learn a specific graph structure. What is the connection between these works and your study? Can you elaborate more about what they miss in their study that you mainly improve?

[1] Ying et al., Do transformers really perform badly for graph representation? Neurips 2021.

[2] Chen et al., NAGphormer: A tokenized graph transformer for node classification in large graphs. ICLR 2023.

[3] Li et al., What Improves the Generalization of Graph Transformers? A Theoretical Dive into the Self-attention and Positional Encoding. ICML 2024.

**Strengths And Weaknesses:**

Strengths:

1. The paper is well-written and easy to follow.

2. The performance of the proposed method is impressive.

3. The proposed method makes sense and follows a simple formulation.

Weaknesses: Please see the requested changes.

---

> ### Author Response · Authors · 2025-02-05
> **Rebuttal to Reviewer mRfw**
>
> Heartfelt thanks are offered to the reviewer for noting the strengths of this work.
> The positive remarks on the clarity, ease of understanding, and impressive performance of the proposed method are greatly appreciated.
> All identified weaknesses will be comprehensively addressed in a detailed, point-by-point format.
>
> $~$
>
> > It is still not clear why the proposed method works better than existing methods, especially on heterophilic graphs. I think the proposed method emphasizes local graph information, which should be contradictory to a successful learning of a heterophilic graph.
>
> All the heterophilic datasets analysed in this paper exhibit a distinct trend of k-hop homophily metric values.
> These trends are demonstrated in Part A of Figure 1 in the paper on page 4.
> The values show an initial increase, then decline towards zero, with the 2-hop neighborhoods exhibiting the highest homophily values.
>
> The k-hop homophily plots essentially show that vertices in the $2$-hop neighbourhood are the most useful for classification in heterophilic graphs.
> *2-hop neighbourhoods are still local* because they retain meaningful label structure around each vertex without requiring full global attention.
> LocalFormer outperforms previous methods because it dynamically selects *the most relevant local information*.
>
>
> $~$
>
> > I am wondering what the performance of the proposed method is on some simple and basic heterophilic graphs, such as Chameleon, Actor, Squirrel, Wisconsin, etc., used in Geom-GCN.
>
> Following the suggestion of the reviewer, experiments have been conducted on six more datasets—three WebKB datasets (Texas, Cornell, Wisconsin) and three Wikipedia datasets (Chameleon, Actor, Squirrel)—to evaluate the performance of our proposed method.
> The WebKB datasets (Texas, Cornell, and Wisconsin) exhibit fluctuating k-hop homophily trends, with values neither consistently decreasing nor following the heterophilic pattern we previously observed (initial increase followed by a decline).
> The Wikipedia datasets (Chameleon, Actor, and Squirrel) present intriguing k-hop homophily patterns, with values fluctuating close to zero across different hop sizes.
>
> We selected the three best-performing transformer baselines (Exphormer, SGFormer, CoBFormer) on the initial three heterophilic datasets as baselines for comparison in this new study.
> We found that our proposed methods (CLF and WLF) performed competitively across all six datasets, though the performance differences were less pronounced than in the three heterophilic datasets initially examined in the paper.
> Please see the updated draft of our paper for more details, specifically Appendix Section "A.5: Experiments on More Non-Homophilic Datasets."
>
>
> $~$
>
> > I think your paper may claim an interesting thing, i.e., the training scheme beyond the architecture itself can further benefit graph learning with Transformers. Existing works [1,2,3] prove that a proper positional encoding is expressive enough or guaranteed to learn a specific graph structure. What is the connection between these works and your study? Can you elaborate more about what they miss in their study that you mainly improve?
>
> While existing works [1,2,3] demonstrate that proper positional encoding can ensure expressivity and capture specific graph structures, they primarily focus on architectural modifications rather than the training process itself.
> Our paper introduces CollaborativeLocalFormer (CLF) and WarmLocalFormer (WLF), which systematically mitigate the over-globalising issue in Graph Transformers—an aspect that prior studies have overlooked.
>
> Specifically, Graphormer [1] enhances expressivity through structural encodings but does not address how Transformers can overemphasize global dependencies. Similarly, NAGphormer [2] proposes tokenization-based neighborhood aggregation for scalability but does not explore training strategies to balance local and global learning. While Li et al. [3] provide theoretical insights into Graph Transformers’ generalization, they do not propose explicit training methodologies to optimize learning dynamics.
>
> Fundamentally, all Graph Transformer architectures inherently suffer from the over-globalising issue due to the nature of the global attention mechanism, which distributes focus across all nodes rather than emphasizing essential local structures [4]. This phenomenon persists regardless of architectural modifications, as global attention mechanisms tend to dilute important local relationships, leading to suboptimal learning.
>
> Our primary objective in this paper is to explicitly mitigate this issue through carefully designed training strategies rather than architectural alterations. The superior performance observed in our experimental results is not merely due to additional model complexity but is instead a direct consequence of effectively alleviating this issue, ensuring a more balanced integration of local and global information in Graph Transformers.

---

> > ### Author Response · Authors · 2025-02-05
> > **Rebuttal to Reviewer mRfw Continued**
> >
> > References:
> > 1. Ying et al., Do transformers really perform badly for graph representation? Neurips 2021,
> > 2. Chen et al., NAGphormer: A tokenized graph transformer for node classification in large graphs. ICLR 2023,
> > 3. Li et al., What Improves the Generalization of Graph Transformers? A Theoretical Dive into the Self-attention and Positional Encoding. ICML 2024,
> > 4. Xing et al., Less is More: on the Over-Globalizing Problem in Graph Transformers, ICML 2024.

---

> > > ### Comment · Reviewer_mRfw · 2025-02-07
> > >
> > > Thank you for the response. The added experiments and discussion look great. In terms of your answer to the first question, I think you mean my understanding is inaccurate, and actually, your model can capture any hop of the information, right? I somehow get it. By the way, I noticed another empirical work of Graph Transformer [1], which might be added as a baseline.
> > >
> > > [1] Luo et al., Neurips 2024. Enhancing Graph Transformers with Hierarchical Distance Structural Encoding.

---

> ### Author Response · Authors · 2025-02-07
> **Interesting Direction for Future Research**
>
> > I noticed another empirical work of Graph Transformer [1], which might be added as a baseline.
>
> Thanks for sharing the paper [1]. The study introduces a structural encoding method, relying on graph coarsening algorithms such as METIS, Spectral, Loukas, Newman, and Louvain. The authors performed thorough experiments, including baseline comparisons, ablation studies, and sensitivity analyses, *primarily on graph classification and long-range datasets where over-globalising is typically less problematic [2]*.
>
> However, for node classification datasets, over-globalising is a significant issue [2] and the datasets we explored differ significantly from those explored in the paper [1]. Incorporating this method as a baseline requires careful hyperparameter tuning and design decisions. These include selecting the most effective coarsening algorithm, the number of levels in the hierarchy, and the number of nodes at each level.
>
> The paper's hierarchical encoding quality largely depends on the graph coarsening method's ability to group nodes effectively. Ineffective coarsening can lead the transformer to misinterpret node distances, resulting in suboptimal attention scores and increased over-globalising. Investigating the issue of over-globalising with hierarchical distance encoding and graph coarsening remains an uncharted area, presenting a valuable opportunity for future research.
>
> $~$
>
> > Actually, your model can capture any hop of the information, right?
>
> Yes, that is correct. The model can capture information from any hop due to the local module's focus on locality. However, it is important to note that if the local module is very deep and applied naively, the local module may face its own challenges, such as over-smoothing.
>
> References:
> 1. Luo et al., Enhancing Graph Transformers with Hierarchical Distance Structural Encoding, NeurIPS 2024,
> 2. Xing et al., Less is More: on the Over-Globalizing Problem in Graph Transformers, ICML 2024.

---

> > ### Comment · Reviewer_mRfw · 2025-02-07
> >
> > Thank you. The response makes sense. I will strongly support this paper if there is any needed discussion later.

---

> > > ### Author Response · Authors · 2025-02-11
> > > **Expressing Gratitude for the Support**
> > >
> > > Thanks a lot for the support. The thoughtful feedback and encouragement are deeply appreciated.

---

### Author Response · Authors · 2025-02-06
**Summary of all Changes Made to the Paper**

Dear all,

The reviewers have recommended improvements to strengthen the submission's contributions. These changes have been integrated into the updated draft, highlighted in blue font. The following list provides a detailed account of all modifications made to the submission:

1. **[Clarifying Mutual Supervision in Collaborative Loss Function]** The concept of mutual supervision within the collaborative loss, along with its mathematical definition, has been described more rigorously. Please refer to the paragraph titled 'Intuition and Details of the Collaborative Loss Function' on Page 7.
2. **[Incorporating Expanded Experiments]** Performance comparisons have been made on six additional datasets—three WebKB (Texas, Cornell, Wisconsin) and three Wikipedia (Chameleon, Actor, Squirrel) datasets—to assess the performance of our proposed methods. The WebKB datasets demonstrated varied k-hop homophily trends as in Figure 9, while the Wikipedia datasets exhibited patterns with values fluctuating around zero as in Figure 10. We compared our method against the top-performing transformer baselines (Exphormer, SGFormer, CoBFormer) from the initial study. Our methods (CLF and WLF) showed competitive performance across all six datasets, shown in Tables 5 and 6. Further details can be accessed in Appendix Section 'A.5: Experiments on More Non-Homophilic Datasets' on pages 19, 20, and 21.
3. **[Addressing Counter-Intuitive Observations and Future Directions]** On page 12, new discussions emphasize exploring sophisticated structural dynamics in heterophilic and non-homophilic datasets, linked to counter-intuitive homophily values, as a promising research direction. Additionally, the Future Work section suggests refining homophily metrics to enhance structural insights, drawing on recent studies to inform this analysis.
4. **[Positioning the Contributions in Relation to Prior Work]** A new paragraph on page 3 positions the paper's contributions to address the overlooked over-globalising issue in Graph Transformers, focusing on training processes rather than architectural changes. It underscores that these strategies enhance performance by effectively balancing local and global information, distinguishing this work from prior studies that have predominantly focused on architectural modifications.
5. **[Justifying NodeFormer's Selection for Attention Visualisation]** On page 5, a new section explains the selection of NodeFormer due to its state-of-the-art design in attention mechanisms for efficiency, making it ideal for analyzing attention score distribution trends. The section highlights that, despite its advanced design, NodeFormer also experiences over-globalising issues, which become apparent in simple attention score visualizations, emphasizing the need for addressing this challenge.
6. **[Justifying Module Selection in our Proposed Framework]** On page 17, a new paragraph links to experiments and ablation studies illustrating that our proposed approaches: WarmLocalFormer (WLF) and CollaborativeLocalFormer (CLF) outperform individually trained GAT and BGA models, as well as the CoBFormer baseline. The linked findings, supported by experiments from both heterophilic and homophilic datasets, validate the effectiveness of integrating the chosen modules — GAT and BGA — in the LocalFormer models.
7. **[Motivating the Homophily Metric]** A motivation section has also been added at the beginning of Section 3.2, right before introducing the formal definition of the homophily metric.
8. **[Correcting Typographical Errors]** All typographical mistakes pointed out by the reviewers have been addressed and rectified.

Once again, sincere gratitude is expressed for the reviewers' assistance in enhancing the contributions of the work.

---

### Decision · Action_Editor_MNc1 · 2025-04-02

**Recommendation:** Accept as is

**Comment:**

The authors have made significant efforts to address all review comments, and all reviewers are satisfied with the revised submission.

**Audience:**

yes

**Claims And Evidence:**

yes